# Peripheral sequestration of huntingtin delays neuronal death and depends on N-terminal ubiquitination
Ayub Boulos[1,3], Dunia Maroun[1], Aaron Ciechanover [2] & Noam E. Ziv [1] ✉

Huntington's disease (HD) is caused by a glutamine repeat expansion in the protein huntingtin. Mutated huntingtin (mHtt) forms aggregates whose impacts on neuronal survival are still debated. Using weeks-long, continual imaging of cortical neurons, we find that mHtt is gradually sequestrated into peripheral, mainly axonal aggregates, concomitant with dramatic reductions in cytosolic mHtt levels and enhanced neuronal survival. in-situ pulse-chase imaging reveals that aggregates continually gain and lose mHtt, in line with these acting as mHtt sinks at equilibrium with cytosolic pools. Mutating two N-terminal lysines found to be ubiquitinated in HD animal models suppresses peripheral aggregate formation and reductions in cytosolic mHtt, promotes nuclear aggregate formation, stabilizes aggregates and leads to pervasive neuronal death. These findings demonstrate the capacity of aggregates formed at peripheral locations to sequester away cytosolic, presumably toxic mHtt forms and support a crucial role for N-terminal ubiquitination in promoting these processes and delaying neuronal death.

Nine known neurodegenerative diseases are caused by genomic expansions of Cytosine-Adenine-Guanine (CAG) repeats in the coding regions of particular proteins, resulting in mutated proteins with elongated stretches of glutamines (polyQ). The best studied disease in this class is Huntington's disease (HD), a progressive, fatal neurological disorder, for which no cure currently exists. HD is caused by a substantial polyQ expansion in the N-terminal domain of Huntingtin (Htt), a very large protein (~3144 amino acids) whose physiological roles are only partly understood[1,2]. This expansion, in turn, results in aggregation prone mHtt variants that ultimately drive severe motor dysfunction as well as cognitive deficits and neuropsychiatric abnormalities[3]. HD symptoms are attributed to the degeneration of neurons in the striatum, yet neurons in the neocortex and other brain regions are also affected, explaining, perhaps, the extensive range of both cognitive and motor symptoms[3–5]. Although HD is relatively rare[6,7], the unequivocal identification of its genetic source has led to a disproportionately large focus on its underlying causes, probably due to commonalties with vastly more prevalent neurological diseases in which the precipitating causes are more elusive. One such commonalty is the presence of abnormal protein aggregates, giving rise to hopes that insights gained on mHtt aggregation will be informative in broader contexts as well.

It is now clear that HD etiology is primarily linked to a 'gain of function' introduced by the aforementioned polyQ expansion[3,6]. In fact, a single copy of mHtt or even of HTT exon 1, which encodes the (expanded) polyQ tract, is sufficient to induce HD symptoms[8–11]. This expansion seems to impair protein folding, resulting in the formation of mHtt oligomers and fibrils[11–14]. These, in turn form large insoluble inclusions both in the nucleus and the cytoplasm (e.g., refs. 15–19). Cells (e.g., refs. 16,18–21), model organisms (e.g., ref. 9) and patients (e.g., refs. 15,22,23) expressing expanded polyQ variants of huntingtin or its fragments generate massive huntingtin-rich inclusions, large enough to be visualized by light microscopy. mHtt cytotoxicity seems to be related to this propensity to aggregate, and numerous, not necessarily mutually exclusive pathways were suggested to take part in the consequential damage and cell death (reviewed in refs. 3,6,11). Paradoxically, aggregates were also suggested to play protective roles, possibly by reducing cytosolic levels of small mHtt oligomers that appear to be the most harmful mHtt forms[20,21,23–27].

Degradation of misfolded proteins is one of the first lines of defense cells have when faced with aggregation-prone, toxic proteins, and the ubiquitin-proteasome system (UPS) is a key degradation mechanism in this regard[28]. Indeed, mHtt aggregates label strongly when probed with antibodies against ubiquitin and are commonly found in tight association with ubiquitin conjugating / deubiquitinating enzymes, ubiquitin-interacting proteins and proteasomes (e.g., refs. 15,16,19,29–33; reviewed in ref. 34). In line with these findings, proteomic comparisons of Htt ubiquitination

[1]Technion Faculty of Medicine, Rappaport Institute and Network Biology Research Laboratories, Fishbach Building, Technion City, Haifa, Israel. [2]Rappaport Faculty of Medicine and Rappaport Technion Integrated Cancer Center (RTICC), Technion-Israel Institute of Technology, Haifa, Israel. [3]Present address: Department of Neurology, Massachusetts General Hospital, and Harvard Medical School, Charlestown, MA, USA. ✉e-mail: noamz@technion.ac.il

profiles in two separate HD animal models revealed that mHtt, but not wild-type Htt, is selectively ubiquitinated on two lysine residues (K6 and K9) at the N-terminus of mHtt in a segment encoded by HTT exon 1[35,36]. This specificity for mHtt agrees with prior observations that WT Htt degradation rates are not affected by suppressing proteasomal activity[37]. Yet, when live imaging was used to follow aggregate formation rates of EGFP-tagged human mHtt exon-1 with a 134 glutamine repeat (Htt134Q:EGFP) in neurons, it was found that the mutation of lysines 6 and 9 to arginines (abolishing ubiquitination at these sites) strongly reduced the rates at which large aggregates appeared[36]. Surprisingly, biochemical approaches revealed that these same mutations increased the insoluble fraction of the fusion protein, probably reflecting minuscule oligomers or 'protofibrils' undetectable by conventional light microscopy[36]. Moreover, when expressed in cell lines, the expression of the fusion protein in which Lysines 6 and 9 were mutated to arginine (Htt134Q(K > R):EGFP) significantly impaired cell viability as compared to Htt134Q:EGFP expression. Of note, lysines 6 and 9 are part of a conserved 17 amino acid N-terminal segment upstream of the polyQ stretch, that plays crucial roles in mHtt oligomerization and fibril nucleation[12]. It was thus suggested that ubiquitin conjugation at these sites promotes the formation of large, visible mHtt aggregates, which, by sequestering away oligomeric or protofibrillar forms of mHtt, decreases proteotoxicity and delays cell death[34,36,38,39]. While this interpretation is congruent with the aforementioned protective roles of aggregates, key questions remain open: What are the temporal relationships between mHtt aggregate formation and changes in cytosolic mHtt levels? Is aggregate formation indeed associated with substantial reductions in cytosolic mHtt levels? Where do these presumed sequestration sites form (nuclei, somata, dendrites, axons)? What is their impact on neuronal survival? Are the molecular dynamics of sequestered mHtt congruent with the suggestions that they effectively 'soak up' and sequester cytosolic mHtt? How are these processes affected by preventing mHtt ubiquitination on lysines 6 and 9?

Here we used tagged variants of the human mHtt N-terminal segment encoded by exon 1, cortical neurons in culture, long-term microscopy and in-situ pulse-chase methods to address these questions. Our findings, described below, provide strong support for the notion that mHtt aggregates, formed at peripheral neuronal locations, act very effectively to sequester away cytosolic mHtt, prevent the formation of nuclear inclusions and greatly enhance the ability of neurons to handle mHtt loads. Moreover, the findings support a crucial role for N-terminal ubiquitination in these sequestration processes.

## Results
### The formation of mHtt aggregates is associated with dramatic reductions in cytosolic mHtt levels
We have previously used cortical neurons in culture and long term imaging (~2 weeks) to record the formation of aggregates of EGFP-tagged human mHtt exon-1 with a 134 glutamine repeat (Htt134Q:EGFP) and in particular, the rates of aggregate appearance and growth kinetics[36]. In those experiments, the morphological features of the cells expressing Htt134Q:EGFP were not visualized. Consequently, cytosolic mHtt levels were not measurable and sites of aggregate formation (axons, dendrites, somata, nuclei) were not resolved. Moreover, Htt134Q:EGFP expression and long-term imaging were initiated approximately at the same time, and thus, the effects of Htt134Q:EGFP expression on cell viability beyond the two-week imaging periods remained unknown. Therefore, to address the relationships between cytosolic mHtt levels and aggregate formation, to determine where aggregates form and how their formation affects cell viability, we created a lentiviral expression vector (Fig. 1A) that encodes for both Htt134Q:EGFP and mCherry, separated by *Thosea asigna* virus 'self-cleaving' peptide[40] (T2A) allowing us to both visualize cell morphology (mCherry) and track mHtt over time (EGFP). In addition, imaging was initiated 8 days after infection in order to collect data in periods that pertain to the aforementioned questions. Specifically, rat cortical neurons were allowed to develop in culture for 12 days. On day 12, neurons were infected with viral particles and 8 days later, the dishes were mounted on a custom-

built confocal microscope equipped with a motorized stage and an automatic focus system. The cell culture dishes were connected to a very slow perfusion system[41], a sterile stream of 5% $CO_2$ / 95% air mixture and heated to ~35–36 °C and spontaneous activity was recorded a measure of overall network viability (See Materials and Methods for further details). Then, long-term imaging was initiated, with image stacks obtained automatically at multiple locations in the dish at 2-hour intervals for three weeks and beyond. This experimental timeline (illustrated in Fig. 1B) allowed us to follow the neurons for a period of about a month from infection.

An example of one such experiment, showing one field of view containing one neuron, is provided in Fig. 1C (see also Supplemental Movie 1). As shown here, both Htt134Q:EGFP and mCherry were initially distributed quite uniformly throughout the somata and dendritic trees. Levels of cytosolic Htt134Q:EGFP initially increased, but with the passage of time, cytosolic Htt134Q:EGFP fluorescence began to diminish substantially; at the same time, large visible aggregates of Htt134Q:EGFP started to appear. Over the next two weeks, aggregate counts increased concomitantly with reductions in cytosolic Htt134Q:EGFP levels (see Fig. 2A for a quantification). In contrast, mCherry expression remained stable and uniform for the entire duration of the experiment (Fig. 2A). Practically identical observations were made in three independent experiments ( = separate cell culture preparations and transductions) and 24 fields of view (FOVs) in total (each FOV reflecting one or more neurons; Fig. 2B). It is worth noting that in these experiments, very few events of cell death were recorded for neurons expressing Htt134Q:EGFP, and that the vast majority of such cells survived even the longest experiments ( > 3 weeks of imaging, or >4 weeks from infection).

These experiments thus suggest that as aggregates begin to form in increasing numbers, they effectively sequester away cytosolic Htt134Q:EGFP, substantially reducing its cytosolic levels and possibly the cytotoxicity associated with cytosolic mHtt (in agreement with prior studies[20,21]).

### Htt134Q:EGFP aggregates form primarily in axons
Prior studies on mHtt aggregation tended to focus on mHtt aggregates or inclusion bodies formed in nuclei or somata of mHtt expressing cells. Somewhat surprisingly, in the experiments described above, few if any aggregates were observed to form within the cell bodies and nuclei of neurons expressing Htt134Q:EGFP. In fact, most of the aggregates appeared to form outside of the mCherry-labeled somatodendritic tree. We also noted, however, that mCherry did not highlight axons very well, in particular remote axonal segments at long distances from cell bodies. Nevertheless, time lapse recordings revealed that many aggregates displayed significant travel in straight lines, in manners suggestive of local transport along axons (Supplemental Movie 2). Taking advantage of the large time series data sets, we carefully examined each aggregate in time and 3D space. Specifically, individual aggregates (labeled with EGFP) were followed forward and backward in time and compared to mCherry labeling at each time point and Z-section. This analysis revealed that most aggregates were associated with faintly labeled axons, as exemplified in Fig. 3A, although for some aggregates, we could not determine the cellular compartment within which they resided (e.g., Fig. 3B). These were therefore classified as 'not resolved'. In addition, a small number of aggregates was associated with dendrites, although we note that this apparent association was often misleading, as such aggregates could move abruptly away from a mCherry labeled dendrite. Notably, we observed no aggregates within cell nuclei at any time-point of these experiments. Such analyses for almost 700 aggregates from three independent experiments and 24 FOVs indicated that the vast majority of Htt134Q:EGFP aggregates were associated with axons (Fig. 3C).

Given the limited utility of mCherry for visualizing distal axons, we resorted to immunolabeling in order to visualize all axons in cortical neuron preparations subjected to the same environmental conditions (slow perfusion, 5% $CO_2$, 35–36 °C). This analysis revealed that ~83% of Htt134Q:EGFP aggregates colocalized with axons immunolabeled against neurofilament heavy chain (380 aggregates from 12 FOVs from three experiments from two cell culture preparations; Supplementary Fig. 1).

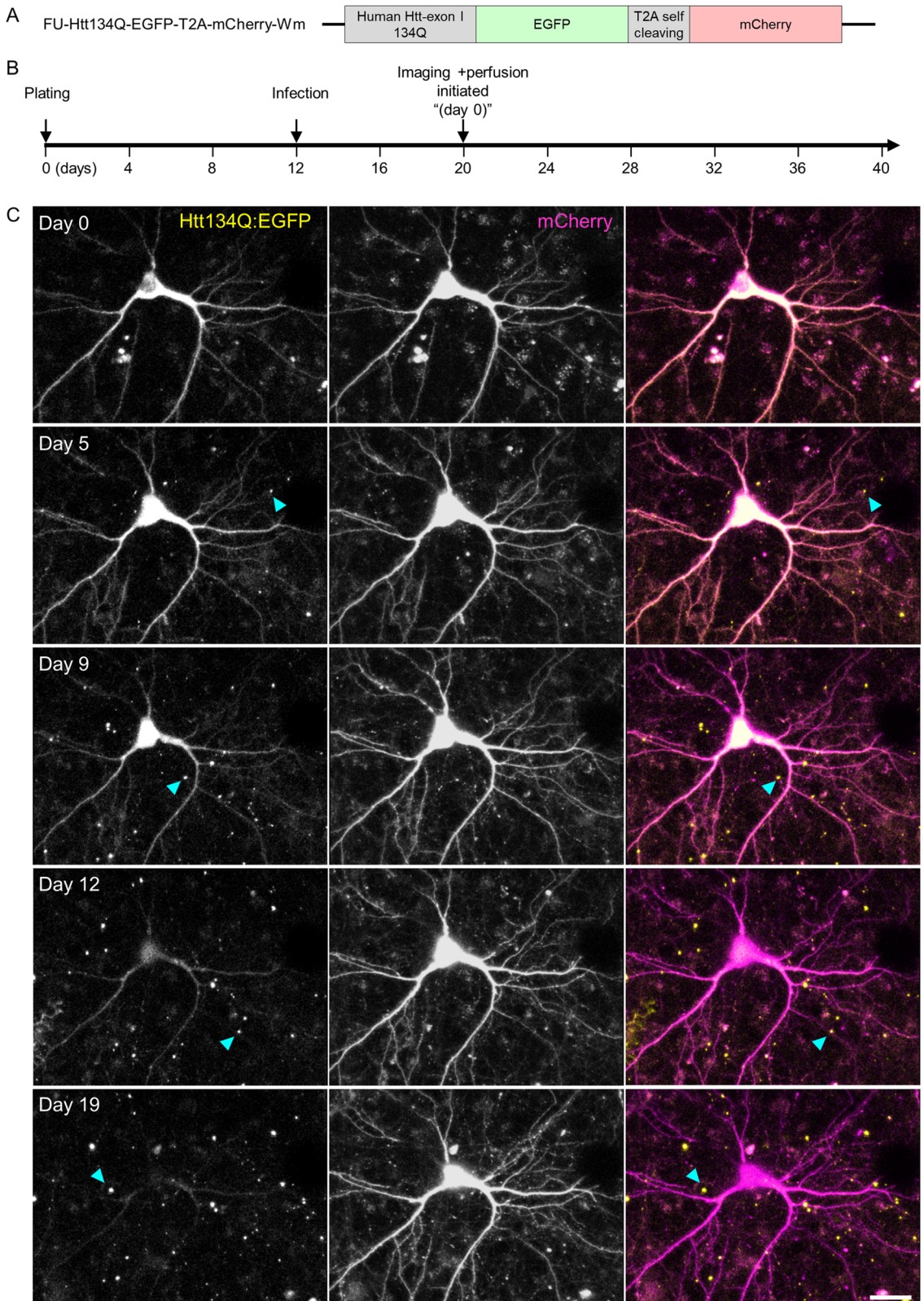

**Fig. 1 | Long-term imaging of neurons expressing Htt134Q:EGFP:T2A:mCherry.**
**A** Schematic illustration of the Htt134Q:EGFP:T2A:mCherry fusion expression vector used in these experiments. **B** Experimental time line. **C** Time-lapse images of cortical neurons expressing Htt134Q:EGFP and mCherry. Times from the initiating of the imaging session indicated at the top left. Cyan arrowheads point to Htt134Q:EGFP aggregates. Scale bar: 20 µm.

Interestingly, in all these experiments, the preparations were perfused slowly with fresh media ( ~ 2 volumes/day). We[41–43] and others[44] have found that under these conditions, neurons in culture are much more viable and resilient to insults such as pharmacological suppression of protein synthesis[45]. Indeed, in a separate set of experiments carried out on neurons maintained in small volumes in an incubator (i.e., without perfusion) and stained against Microtubule Associated Protein 2 (MAP2; somatodendritic compartments) and Neurofilament-heavy chain (axons), a shift toward a

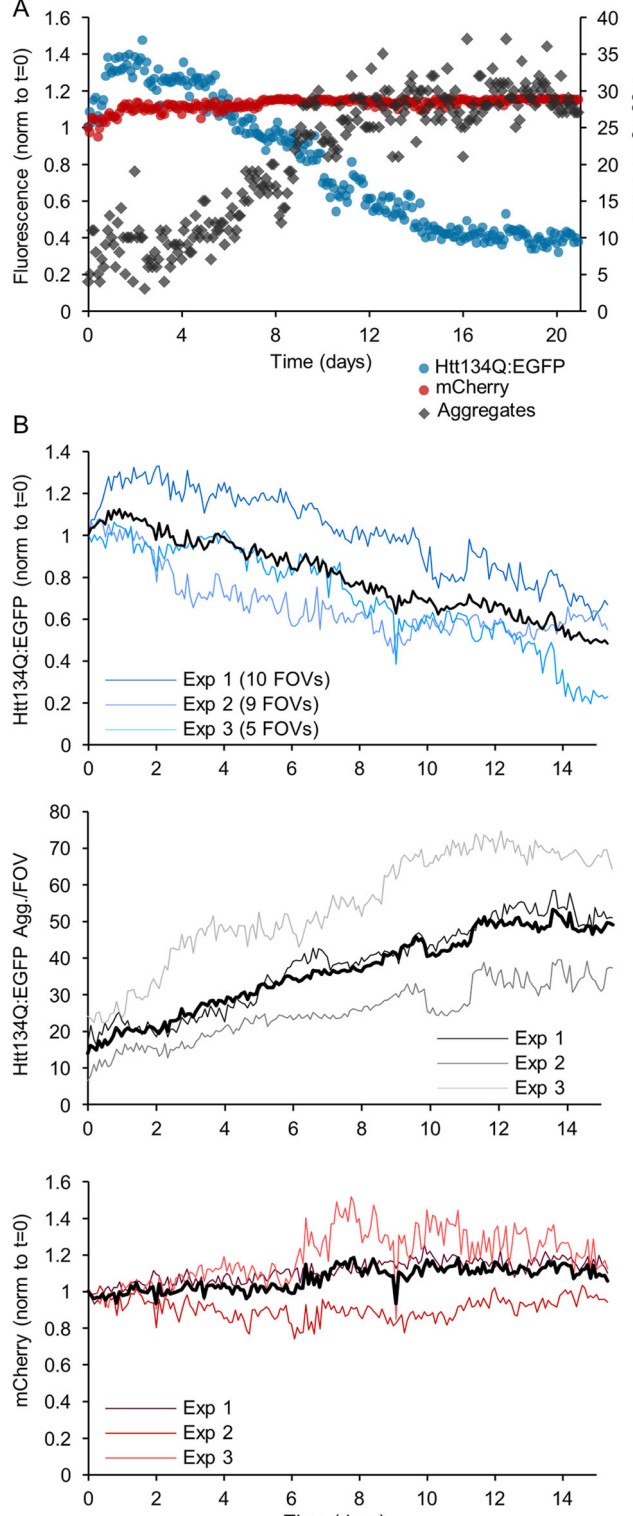

**Fig. 2 | mHtt appearance is associated with the disappearance of cytosolic Htt134Q:EGFP. A** Changes in Htt134Q:EGFP cytosolic fluorescence, cytosolic mCherry fluorescence and Htt134Q:EGFP aggregate counts in the field of view shown in Fig. 1C. Fluorescence was normalized to values at t = 0. Day 0 indicates the first day of imaging (8 days post-transduction; see Fig. 1B). **B** Average changes in the fluorescence of cytosolic Htt134Q:EGFP and mCherry, and in Htt134Q:EGFP aggregate counts in three independent experiments. Each line is the average from one experiment for all Fields of view (FOVs) each of which contains at least one neuron. Numbers of FOV for each experiment indicated in top panel. Black lines represent the average for all three experiments. All fluorescence values were normalized to their initial values at t = 0.

somatodendritic localization was observed, including aggregates within nuclei (Supplementary Fig. 2).

Taken together, these findings suggest that following an initial increase in cytosolic mHtt concentrations, large, visible mHtt aggregates begin to form, preferably at peripheral neuronal locations (mainly axons), and this is followed by strong reductions in cytosolic mHtt concentrations.

## Aggregate-associated mHtt is continuously exchanged with cytosolic and newly synthesized mHtt

The data so far are consistent with the possibility that cytosolic mHtt is sequestered into large, visible and mainly axonal aggregates, and that this sequestration leads to dramatic reductions in cytosolic mHtt concentrations. In this regard, the fact that individual aggregates do not grow indefinitely might suggest that aggregates ultimately attain a form of equilibrium with cytosolic mHtt, that is, rates of mHtt association with- and dissociation from aggregates become similar. Alternatively, this might be explained by reductions in mHtt synthesis coupled to the formation of 'permanent mHtt deposits', in which mHtt, once incorporated, is practically never released.

To obtain a better understanding of the nature of mHtt aggregates, their capacity to act as cytosolic mHtt 'sinks', the molecular dynamics of aggregate-associated mHtt and to disambiguate the two aforementioned interpretations, we carried out in-situ pulse-chase experiments using HaloTag technology[46]. HaloTag protein is a modified bacterial monomeric haloalkane dehalogenase to which HaloTag ligands bind covalently via a reactive haloalkane linker. Non-toxic, fluorescent HaloTag ligands can then be used to rapidly label HaloTag fusion proteins, which are otherwise non-fluorescent. By saturating the HaloTag protein with one label and then exposing cells to a second, spectrally separable label, newly synthesized protein copies can be visualized separately from preexisting protein copies (e.g., refs. [47–49]). To that end, we constructed a lentiviral expression vector encoding for a fusion protein of Htt134Q and HaloTag protein (Htt134Q:HaloTag; Fig. 4A). We then carried out the experiments outlined in Fig. 4B. Briefly, cortical neurons were infected with the Htt134Q:HaloTag-encoding lentiviral vectors, and 8 days later, mounted on the imaging system described above, connected to the slow perfusion system and a 5% $CO_2$ stream. On day 22 in culture, a first HaloTag ligand (Oregon green) was added to the dishes for one hour to label the preexisting mHtt134Q:HaloTag pool. The cells were then washed and exposed to saturating concentrations of a non-fluorescent HaloTag ligand[49] (CPXH) to block residual HaloTag binding sites, and imaging was initiated. After 4 hours, the cells were washed again, and a second HaloTag ligand (JF635-HT) was added to the media. Imaging was then continued at 1.0 to 1.5-hour intervals as described above, focusing on Oregon Green-labeled mHtt134Q:HaloTag aggregates.

One such experiment is illustrated in Figs. 4C and 5A–C. In this experiment as well as in all others (Fig. 5D, E; 3 independent experiments, 53 fields of view in total), we found that the fluorescence intensity of the first ligand (Oregon Green) at individual aggregates initially increased over the first ~30 hours, presumably reflecting the sequestration of preexisting cytosolic mHtt134Q:HaloTag protein copies. This was followed by a gradual reduction in the fluorescence of this first ligand (Fig. 5A, B, D). In parallel, the fluorescence intensity of the second ligand (JF635-HT) at the same aggregates (as well as at new aggregates), presumably representing newly synthesized copies of mHtt134Q:HaloTag, increased steadily over time (Fig. 5A, C, E). Interestingly, the kinetics of this increase were remarkably similar to those of Htt134Q:EGFP aggregation kinetics (Fig. 5E, gray line) measured at newly appearing Htt134Q:EGFP aggregates in the experiments described in Figs. 1–3.

Prior studies[50,51] as well as our own observations that not all Htt134Q:EGFP aggregates colocalized with discernable mCherry-filled axons or dendrites (Fig. 3) raised the possibility that some mHtt aggregates are in fact extracellular mHtt deposits. Although this possibility is not congruent with the gradual replacement of preexisting mHtt134Q:HaloTag copies with new ones, we repeated the experiments described above but here, instead of blocking residual unoccupied HaloTag sites with CPXH, we added the membrane impermeable fluorescent HaloTag ligand JF635i-HT

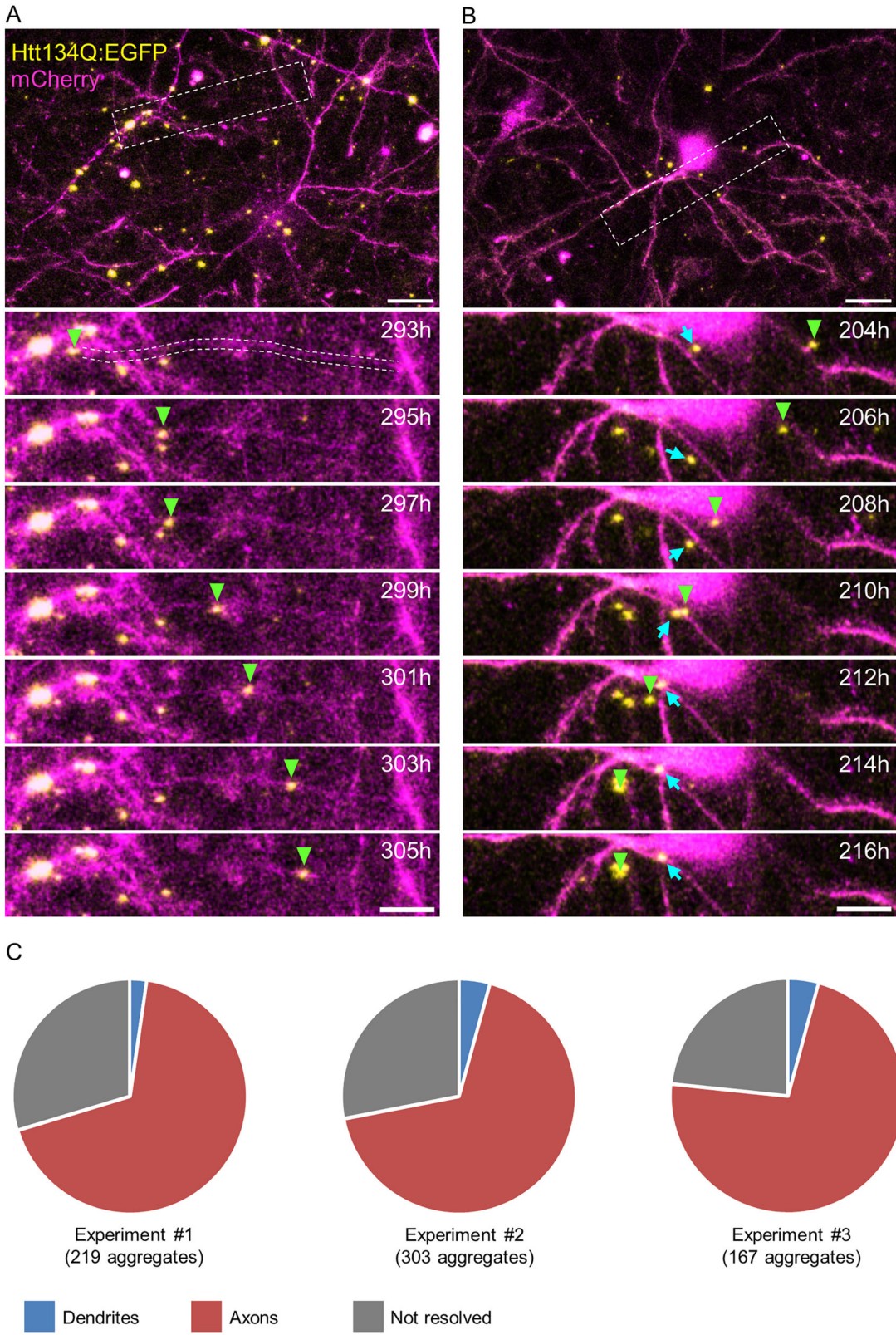

**Fig. 3 | Analyses of mHtt aggregate localization. A** Time-lapse series of a Htt134Q:EGFP aggregate (indicated by the arrowhead) which appears to be moving along a faintly visible axon. A low magnification of the field of view is shown in the top panel. **B** Time-lapse series of a Htt134Q:EGFP aggregate (indicated by the arrowhead) assigned to the 'unresolved' category. A low magnification view of this field of view is shown in the top panel. Times in hours indicated at top right. Scale bars: top panels 20 μm, bottom panels 10 μm. **C** Aggregate assignment to different neuronal compartments. Three different experiments, aggregate numbers from each experiment indicated below each pie chart.

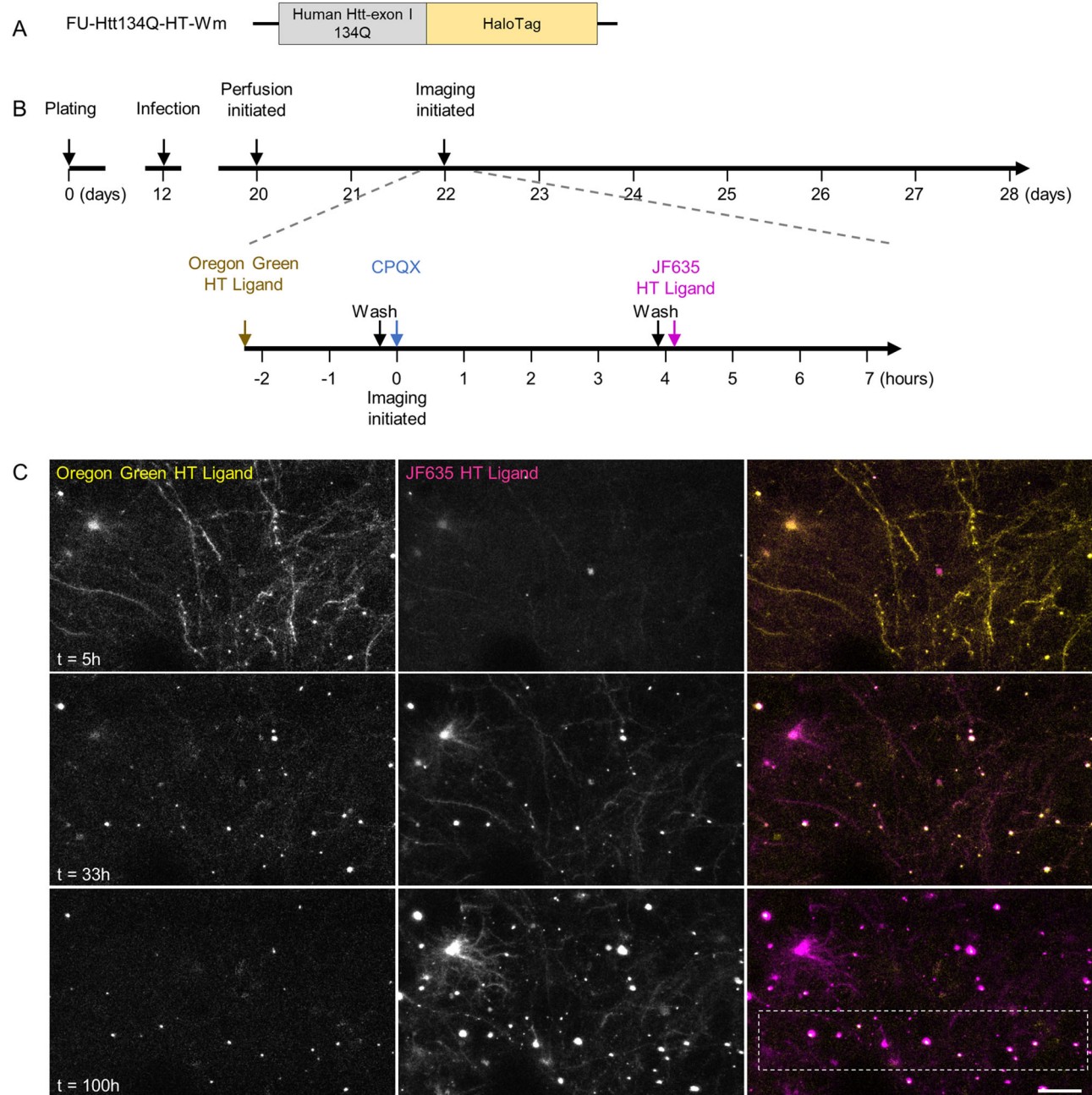

**Fig. 4 | Pulse-chase measurements of mHtt turnover at individual Htt134Q:HaloTag aggregates. A** Schematic illustration of the Htt134Q:HaloTag fusion protein. **B** Illustration of the experiment timeline and the pulse-chase procedure. **C** Time-lapse images of cortical neurons expressing Htt134Q:HaloTag after labeling with the first HaloTag ligand (Oregon green HaloTag) and the second ligand (JF635-HT). Times after the first labeling indicated at bottom left. Area enclosed in dashed line rectangle is shown at higher resolution in Fig. 5. Scale bar: 20 μm.

to the media, assuming that if mHtt aggregates are in fact extracellular deposits, they should label with this extracellular ligand. However, and in line with the findings described above, no aggregate labeling was observed when this membrane impermeable HaloTag ligand was used (Supplementary Fig. 3). Conversely, adding the membrane permeable ligand JF635-HT a few hours later, resulted in the labeling of these same mHtt134Q:HaloTag aggregates as expected.

The pulse-chase described in Figs. 4, 5 indicated that the exchange of aggregate-associated mHtt with cytosolic mHtt pools occurs rather slowly, on time scales of days. However, time scales obtained for mHtt exchange in such experiments are confounded by the time scales over which pools of Oregon Green- (preexisting) and JF635-HT- (newly synthesized) labeled mHtt134Q shrink and grow, respectively. To examine if exchange

occurs also on shorter time scales we carried out fluorescence recovery after photobleaching (FRAP) experiments, using intense illumination to selectively photobleach individual mHtt:134Q:EGFP aggregates, followed by time lapse imaging of fluorescence recovery at the same aggregates (Supplementary Fig. 4A,B). Analyses of ~35 aggregates from three independent experiments revealed only limited recovery over time scales of about 5 hours (time constant ≈ 9 h; Supplementary Fig. 4C), confirming that the exchange of aggregate-associated and cytosolic mHtt is relatively slow.

These observations thus strongly indicate that mHtt134Q aggregates are not 'permanent deposits' but pools of sequestered mHtt that slowly exchange their contents with cytosolic mHtt and exchange 'old' mHtt copies with newly synthesized ones. Moreover, these observations negate the possibility that reductions in cytosolic Htt134Q:EGFP levels observed in the

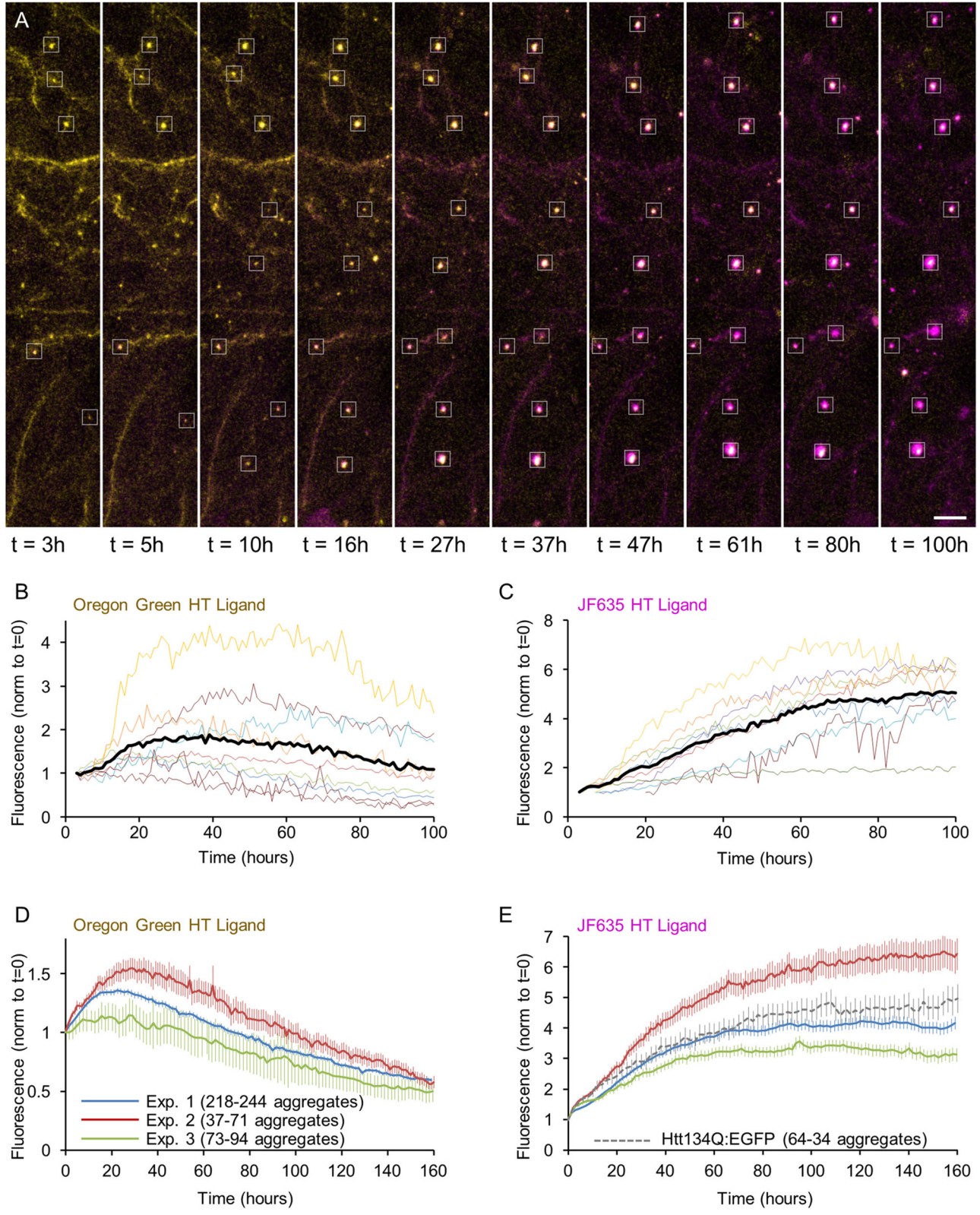

experiments of Figs. 1–3 merely reflect strong reductions in Htt134Q synthesis. In sum, the findings suggest that mHtt aggregates exhibit molecular dynamics congruent with effective sequestration of cytosolic mHtt into extremely concentrated, condensed intracellular mHtt pools which are at equilibrium with cytosolic pools of the same molecule.

## Mutation of mHtt K6 and K9 disrupts aggregate formation and sequestration of cytosolic mHtt

We previously reported[36] that substitution of Htt134Q:EGFP lysine 6 and 9 with arginines, which precludes ubiquitination at these sites, delays the formation and reduces the counts of large and visible Htt134Q:EGFP aggregates

**Fig. 5 | Turnover of aggregate-associated mHtt. A** Time lapse images of individual Htt134Q:HaloTag aggregates demonstrating the transition from yellow (Htt134Q:HaloTag molecules labeled with the first ligand) to magenta (presumably newly synthesized Htt134Q:HaloTag molecules labeled with the second ligand). Boxes indicate aggregates whose fluorescence is quantified in B and C. Scale bar:10 μm. **B** Changes in fluorescence intensities of Oregon Green HaloTag ligand for the 9 aggregates marked with boxes in A. Fluorescence values were normalized to values at t = 0. Thick black line is the average for all 9 aggregates. **C** Changes in the fluorescence intensities of JF635-HT for the aggregates boxed in A. Fluorescence values were normalized to values at t = 0. Thick black line is the average for all 9

boxes. **D** Average changes in the fluorescence intensity of Oregon Green HaloTag ligand of aggregates followed in three independent experiments. Fluorescence values were normalized to values at t = 0. **E** Average changes in the fluorescence intensity of JF635-HT within the same aggregates of D. Fluorescence values normalized to values at t = 0. The gray line depicts the average kinetics of aggregate growth measured for Htt134Q:EGFP in the experiments shown in Figs. 1–3. Data for tracked aggregates were aligned temporally to the moment of their appearance, their growth kinetics were measured and the data was pooled. See ref. 36 for further details on this procedure. Bars, SEM.

while concomitantly increasing the prevalence of very small, insoluble forms of the same protein. The effects of this manipulation on the capacity of the remaining aggregates to sequester cytosolic mHtt remained unknown as did its impact on aggregate localization and neuronal viability. To address these questions, we mutated mHtt lysine 6 and 9 of the fusion protein of Fig. 1A to arginine, resulting in Htt134Q(K > R):EGFP:T2A:mCherry (Fig. 6A, B). We then carried out experiments identical to those described in Figs. 1–3 using this fusion protein except that here imaging (but not mounting on the imaging system) was delayed by 2 days to compensate for the slower appearance of Htt134Q(K > R):EGFP aggregates (Fig. 6C).

One such experiment is shown in Fig. 6D. In this experiment as well as in 4 independent experiments (49 fields of view), and in line with our prior observations, significantly fewer Htt134Q(K > R):EGFP aggregates were observed to form, with total counts remaining much lower (about 5 per FOV compared to about 30 at day 6 of the experiments and about 45 at day 12; Fig. 7B, compare with Fig. 2B). As observed for mHtt134Q:EGFP, cytosolic Htt134Q(K > R):EGFP levels initially increased, but unlike what was observed for mHtt134Q:EGFP, Htt134Q(K > R):EGFP levels remained elevated, exhibiting only modest decreases of even increases over time (Fig. 7A, B; compare with Fig. 2A, B). mCherry fluorescence levels did not exhibit consistent trends, and remained stable on average (Fig. 7B).

These observations indicate that the substitution of lysines 6 and 9 with arginine, which precludes the covalent attachment of ubiquitin chains at these sites, strongly impairs the sequestration away of cytosolic mHtt, resulting in greater cytosolic concentrations of potentially harmful mHtt forms.

### Mutation of mHtt K6 and K9 strongly impairs neuronal viability

We previously reported[36] that Htt134Q(K > R):EGFP (in which lysine 6 and 9 were mutated to arginine) strongly impaired cell viability when expressed in cell lines but effects on neuronal viability remained unknown. As mentioned above, we observed little, if any cell death when following neurons expressing Htt134Q:EGFP even after up to 25 days of continuous imaging. In striking contrast, neurons expressing Htt134Q(K > R):EGFP (K6,9 mutated to arginine) rarely survived beyond day 9 of the experiments, and most of these were observed to die within the first week or so (see Fig. 6D for two examples).

We wondered whether the differential effects on aggregate formation rates might be due to differences in viral titers. We thus infected HEK-293 cells with identical volumes of viral stocks and compared Htt134Q:EGFP / Htt134Q(K > R):EGFP and mCherry fluorescence at days 1,2 and 3 post infection. Differences were minor (Supplementary Fig. 4; if anything, infection was more efficient for Htt134Q(K > R):EGFP:T2A:mCherry) and slightly adjusting infection protocols to accommodate for these small differences had no effect on outcomes.

These observations are thus in line with the possibility that precluding ubiquitination of mHtt lysine 6 and 9 impairs mHtt sequestration and impairs the capacity of neurons to cope with the damage caused by harmful, cytosolic mHtt forms.

### Mutation of mHtt K6 and K9 is associated with the appearance of numerous nuclear aggregates

The substitution of K6 and K9 with arginine resulted in dramatically lower counts of discernable Htt134Q(K > R):EGFP aggregates. Yet, examination of

their cellular localization led to an interesting observation: In preparations subjected to slow perfusion, mHtt134Q:EGFP aggregates were practically never found within the nuclei of neurons expressing this fusion protein. Here, however, bright Htt134Q(K > R):EGFP aggregates were observed in the nuclei of many neurons expressing the mutated fusion protein (see Figs. 6D and 8A for some examples). In fact, nearly half of Htt134Q(K > R):EGFP aggregates were observed to form within neuronal nuclei (Fig. 8B). A similar bias, although not as striking, was observed in preparations that were fixed and stained with antibodies as described above without subjecting them first to slow perfusion periods (Supplementary Fig. 2).

### Mutation of K6 and K9 suppresses molecular dynamics of aggregate-associated mHtt

The lack of evidence for effective mHtt sequestration in neurons expressing Htt134Q(K > R):EGFP led us to examine if this could be explained in part by the properties of Htt134Q(K > R):EGFP aggregates and in particular the molecular dynamics they exhibit. To that end, we mutated lysine 6 and 9 of Htt134Q:HaloTag, obtaining Htt134Q(K > R):HaloTag (Fig. 9A). We then repeated the experiments described in Fig. 4 using this fusion protein instead. As expected for Htt134Q(K > R), fewer aggregates were found following labeling with the first HaloTag ligand (JF479-HT or Oregon green). Over time, labeling with the first label diminished to a variable degree. However, unlike what was observed for Htt134Q:HaloTag, gradual labeling of these same aggregates with the second ligand (JF635-HT) was only rarely observed (Fig. 9B, C), with fluorescence remaining at near background levels. Similar results were observed in three independent experiments (116 aggregates in total).

These findings could be interpreted in two manners: (1) Aggregates of mHtt in which lysine 6 and 9 were mutated to arginine do not exchange mHtt with cytosolic pools or with newly synthesized mHtt copies, and are thus effectively permanent mHtt deposits with limited capacity to buffer excessive cytosolic mHtt; (2) Protein synthesis is impaired in neurons expressing Htt134Q(K > R):HaloTag resulting in little production of new Htt134Q(K > R):HaloTag copies. These interpretations are not mutually exclusive, yet we note that labeling of cytosolic Htt134Q(K > R):HaloTag with JF635-HT was observed in some cells (e.g., Fig. 9B) as was the occasional appearance of new aggregates labeled exclusively with JF635-HT. These observations argue against a complete shutdown of Htt134Q(K > R):HaloTag synthesis, although they do not rule out impairments in protein synthesis in Htt134Q(K > R):HaloTag expressing neurons, in particular in the days preceding their death. The experiments do rule out, however, the possibility that the replacement of Oregon green (or JF479-HT) labeled mHtt with JF635-HT-labeled mHtt (Figs. 4, 5, 9) merely reflects spontaneous dissociation of the first ligand from aggregate-associated mHtt followed by binding of the second ligand to the same mHtt molecules, as this would have been expected to occur for both Htt134Q:HaloTag and Htt134Q(K > R):HaloTag. As no such replacement was observed for Htt134Q(K > R):HaloTag (compare Figs. 5 and 9), this possibility is highly unlikely.

## Discussion

The presence of large protein aggregates in many neurodegenerative conditions has, unsurprisingly, attracted enormous attention and speculation as to their roles in disease etiology, and yet, to this day, important questions

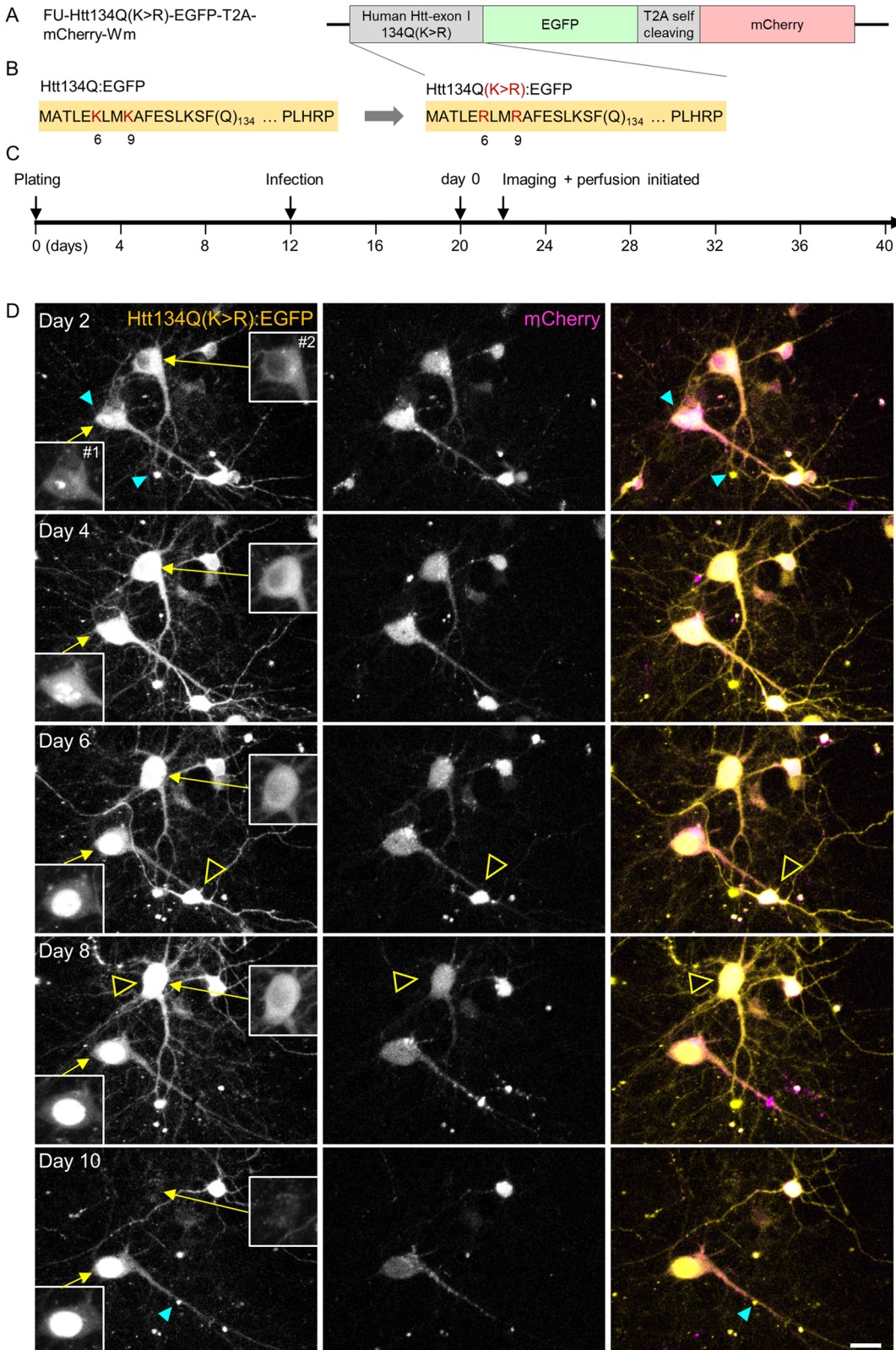

**Fig. 6 | Long-term imaging of neurons expressing Htt134Q(K > R):
EGFP:T2A:mCherry. A** Schematic illustration of the Htt134Q(K > R):
EGFP:T2A:mCherry fusion expression vector used in these experiments.
**B** Illustrations showing the two lysines mutated to arginine. **C** Schematic illustration
of the experiments time line. **D** Time-lapse series of cortical neurons expressing
Htt134Q(K > R):EGFP:T2A:mCherry. Insets show the cell bodies of two neurons at
lower contrast settings. Note the appearance of nuclear aggregates in the bottom cell
(#1) and their merging into a huge nuclear inclusion. Cyan arrowheads point to
aggregates. Open yellow arrowheads point to neurons that died during the experi-
ment. Scale bar: 20 µm.

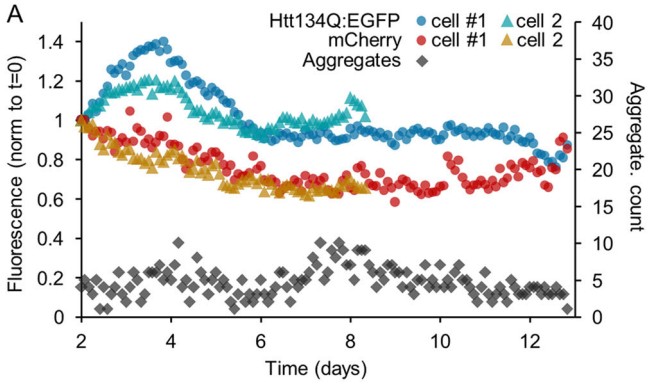

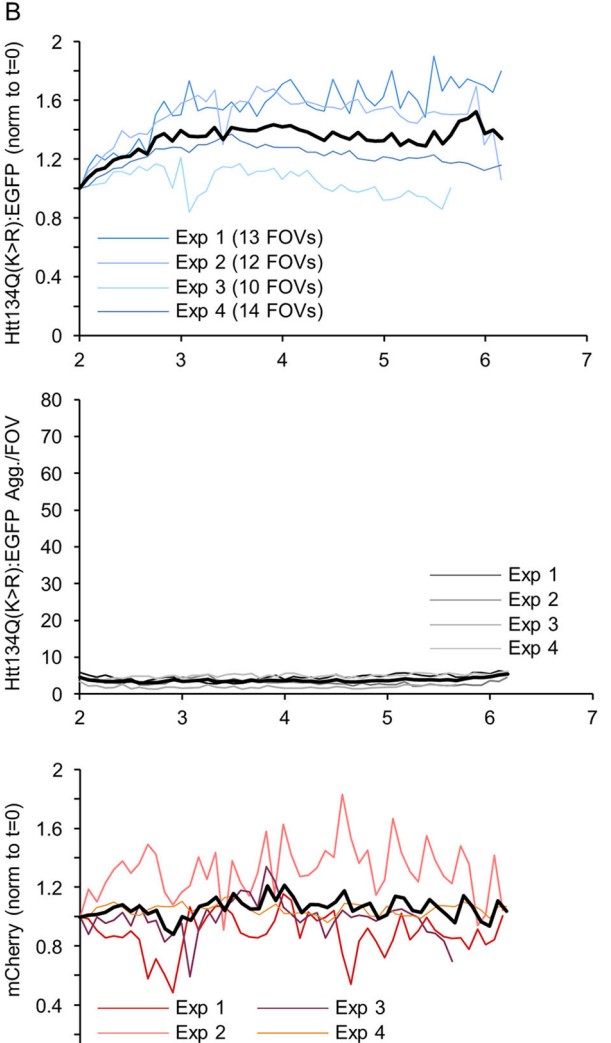

**Fig. 7 | Substitution of K6 and K9 with arginines disrupts aggregate formation and mHtt sequestration. A** Average changes in Htt134Q(K > R):EGFP cytosolic fluorescence, Htt134Q(K > R):EGFP aggregate counts and mCherry fluorescence in two of the neurons shown in Fig. 6D. Fluorescence was normalized to values at t = 0. Day 0 is day 10 post-transduction (see Fig. 6C). **B** Changes in the fluorescence of cytosolic Htt134Q(K > R):EGFP, aggregate counts and mCherry fluorescence. Each line is the average for cells in one of four separate experiments (number of fields of view in each experiment indicated in top panel). Thick black line is the average for all experiments.

regarding such aggregates remain open: Are these major causative factors in disease etiology or, conversely, cellular mechanisms that allow neurons to cope with toxic proteins and delay their cumulative damage? Does their formation reflect a failure of cellular protein degradation mechanisms, or, perhaps, a repurposing of such mechanisms? Here we used tagged variants of the human N-terminal mHtt segment encoded by exon 1 containing an expanded poly-Q stretch, cortical neurons in culture, long-term microscopy and pulse-chase methods to examine relationships between mHtt aggregate formation, cytosolic mHtt levels, aggregate properties and cell death and examine how all these are affected by preventing mHtt ubiquitination at N-terminal lysine residues previously shown to be specifically ubiquitinated in HD animal models. These experiments revealed that initial elevations in cytosolic mHtt levels are followed by the appearance of increasingly greater numbers of mHtt aggregates (Figs. 1, 2), formed primarily at remote axonal locations (Fig. 3, Supplementary Fig. 1), and a concomitant, dramatic reduction in cytosolic mHtt levels (Figs. 1, 2). Pulse-chase (Figs. 4, 5) and FRAP experiments (Supplementary Fig. 4) revealed that mHtt aggregates are dynamic complexes that continually gain newly synthesized mHtt copies and lose older copies over time scales of days, through slow exchange of aggregate-associated and cytosolic mHtt, suggesting that mHtt aggregates function effectively as spatially-confined, highly concentrated mHtt sinks at equilibrium with cytosolic pools of the same protein. In contrast, when lysines 6,9 were replaced with arginines, precluding ubiquitination at these sites, significantly fewer aggregates were observed to form, cytosolic mHtt levels were not reduced effectively (Figs. 6, 7), nuclear mHtt inclusion bodies appeared in many neurons (Fig. 8) and cell death was the rule rather than the exception. Moreover, pulse-chase experiments revealed little evidence for exchange of aggregate-associated mHtt with newly synthesized copies of the same protein (Fig. 9), suggesting that the physiochemical properties of these aggregates were altered, such that they acted, in effect, as permanent mHtt deposits. These findings, strongly support the notion that aggregates can act very effectively to sequester cytosolic mHtt into focal, highly concentrated pools at peripheral cellular sites and by doing so, enhance neuronal survival when faced with potentially lethal loads of toxic mHtt forms. Moreover, the findings support a crucial role for N-terminal ubiquitination in promoting these protective sequestration processes, minimizing the accumulation of cytosolic toxic and insoluble mHtt forms, the formation of nuclear inclusion bodies and ultimately cell death.

All findings reported in this study were based on the expression of the mHtt N-terminal fragment encoded by exon 1 (fused to a reporter), and not of the naïve, full-length protein. This fragment, however, harbors the expanded poly-Q stretch whose length dictates the degree of mHtt pathogenicity[6]. As mentioned in the Introduction, the expression of this segment is sufficient to induce HD symptoms in animal models (e.g., refs. 8–10, reviewed in refs. 11,52). Yet, shorter N-terminal mHtt fragments can be more aggregation-prone than full-length mHtt[34] and tagging with EGFP or HaloTag protein might add a layer of non-physiological effects (e.g., refs. 19; see also ref. 18). Moreover, the EGFP variants were encoded in-line with mCherry separated by a T2A 'self-cleaving' sequence, which results in efficient, but not perfect separation of the two fusion proteins. Finally, in humans (and many animal models), HD develops over years and decades whereas in our experiments, mHtt dynamics were studied over a month at most. These are, unfortunately, important differences that call for some caution in interpreting our findings. Yet, it remains conceivable that aggregation prone fragments with exaggerated lengths of poly-Q tracts (as compared to clinical presentations[6]) capture cellular events associated with HD, albeit on highly compressed time scales.

Beyond the practical aspects of shortening experimental time scales, the experimental system used here offers several important advantages in comparison to other commonly used ex-vivo systems. First, data were collected in neurons rather than in cell lines (e.g., ref. 53) or yeast (e.g., refs. 53–55). Second, the neurons were followed continuously for weeks at relatively short time intervals, allowing for longitudinal measurements at the level of individual aggregates, cellular compartments and neurons. Third, the use of a slow, continuous perfusion system improved cell viability,

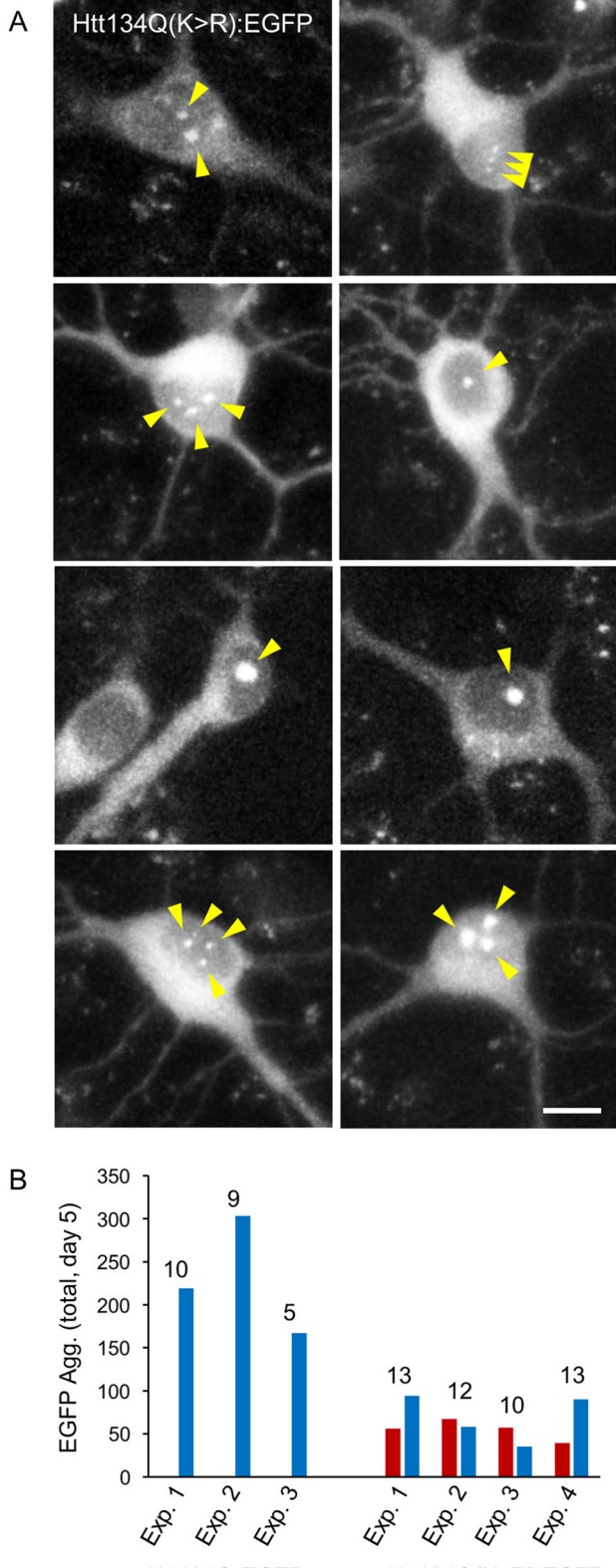

**Fig. 8 | Htt134Q(K > R):EGFP aggregates form in neuronal nuclei.**
**A** Representative images of eight neurons expressing Htt134Q(K > R):EGFP in which nuclear aggregates were observed to form. Scale bar: 10um. No examples of neurons expressing Htt134Q:EGFP are shown as none were found. **B** Comparison of aggregate localization (nuclear - red / extranuclear - blue) in neurons expressing Htt134Q:EGFP:T2A:mCherry and Htt134Q(K > R):EGFP:T2A:mCherry (experiment day 5, numbers indicate the number of fields of view analyzed in each experiment). Bars indicate the total number of aggregates observed in all FOVs of each experiment.

resulted in stable conditions (temperature, osmotic pressure, media composition), eliminating stress and perturbations associated with manual feeding and moving preparations back and forth between imaging systems and incubators (e.g., refs. 20,21), and resulted in aggregate distributions that more closely resemble those observed in HD patients (see below). Thus, while this system does not truly recapitulate in-vivo conditions, it would seem to be advantageous in important aspects.

A final matter concerns our interpretation of the experiments using mHtt fusion proteins in which K6,9 were substituted with arginines. We attributed the observed effects to the preclusion of ubiquitination at these lysines, yet we cannot rule out the possibility that these were related to the presence of the arginines themselves or to other posttranslational modifications at these sites. Indeed, arginine methylation is emerging as an important posttranslational modification in neurodegenerative diseases. Yet arginine methylation at multiple downstream sites was shown improve mHtt solubility and reduce mHtt toxicity[56,57]. This being so, mutating lysines 6 and 9 to arginines would be expected to improve mHtt solubility and reduce cell death, which is the opposite of what was observed here. Lysines 6, 9 were also shown to undergo other posttranslational modifications, including SUMOlation[58] (small ubiquitin like modifier) and acetylation[59] (reviewed in ref. 60), and we cannot rule out the potential roles of these modifications. Yet, the selective presence of ubiquitin on K6,9 in both rat[36] and mouse[35] transgenic HD models, as well as mHtt134Q:EGFP but not Htt134Q(K > R):EGFP[36] point to the preclusion of K6,9 ubiquitination as the most likely explanation for these observations.

The process by which cytosolic mHtt gives rise to large visible aggregates has been studied intensively. A major factor is this process in the nucleation stage, which is followed by aggregate growth, possibly followed by fragmentation and the formation of secondary nuclei (reviewed in ref. 61). Overall rates of nuclei formation depend on intrinsic probabilities of primary nucleation events as well as concentrations of cytosolic mHtt, explaining perhaps the appearance of aggregates only after substantial levels of cytosolic mHtt levels are reached (Fig. 2; see also ref. 55). Interestingly, the N-terminal segment of mHtt, and in particular its first 17 amino acids (which include lysine 6 and 9) and the posttranslational modifications it undergoes has been shown to play crucial roles in fibril nucleation (reviewed in ref. 12). This might explain the slower appearance of visible mHtt fusion proteins aggregates in which lysines 6 and 9 were mutated. Moreover, it is quite easy to imagine how the addition of ubiquitin side-chains to these N-terminal lysines would interfere with the orderly organization proposed for mHtt exon 1 protofibrils[13,14], and how their absence would result in their greater stability, in line with our pulse-chase experiments using mHtt134Q(K > R) (Fig. 9).

In recent years, aggregation processes have been cast into a new framework, suggesting that mHtt aggregates, or inclusion bodies, are dynamic, phase-separated compartments and that the sequestration process, at least initially, reflects a process of liquid-liquid phase separation[53–55]. Our data is, in part, congruent with this view – the gradual appearance of spatially restricted, highly concentrated mHtt pools in apparent equilibrium with dilute cytosolic mHtt pools, and the occasional merging of small aggregates into larger ones (Figs. 1, 2, 4, 5, Supplementary Fig. 3) as might be expected for phase separated mHtt pools. We note, however, that the rates of exchange between pools recorded here (many hours to days) were slower those recorded in non-neuronal cells in some of the aforementioned studies (minutes to a few hours[53–55]) suggesting that condensed mHtt clusters are relatively solid (see also refs. 18,19,53). It is also somewhat puzzling that aggregates appeared initially at remote axonal locations, where presumably, mHtt concentrations would be lower than those in the vicinity of major somatic protein synthesis centers. In this regard it is worth mentioning another line of work on the biophysics of aggregate formation that highlights the importance of collisions between mHtt molecules for aggregate nucleation and growth (e.g., refs. 55,62,63). It is conceivable that the highly constrained volumes of axons combined with the cytosolic 'mixing' associated with extensive transport processes within these confined volumes would greatly increase the likelihood of such collisions and preferably drive aggregate nucleation and growth at these remote locations. Along these

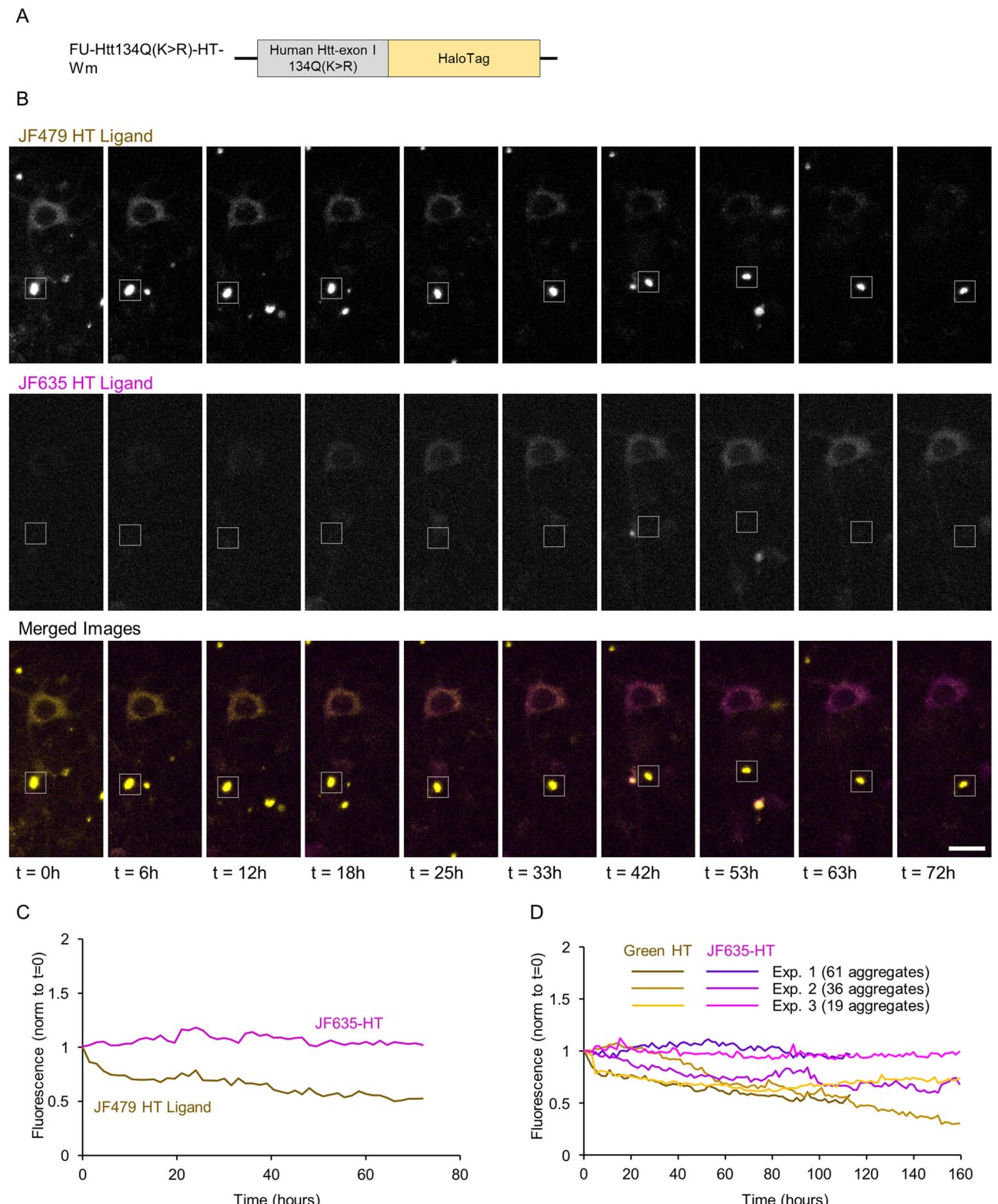

**Fig. 9 | Pulse-chase measurements of mHtt turnover at individual Htt134Q(K > R):HaloTag aggregates. A** Schematic illustration of the Htt134Q(K > R):HaloTag fusion protein used in these experiments. **B** Time lapse images of aggregate-associated mHtt turnover, which should manifest as a transition from yellow (first ligand) to magenta (second ligand). Box indicates an example of a tracked aggregate. No discernable transition is observed for this aggregate in spite of the gradual transition observed in the cell body of a nearby neuron (above box). Scale bar: 20 μm. **C** Quantification of Oregon-Green HT and JF635-HT for the aggregate shown in B. **D** Average changes in the fluorescence intensities of JF479 / Oregon green and JF635-HT for a total of 116 aggregates from three independent experiments. Fluorescence values normalized to values at t = 0.

lines, once formed, many aggregates, in particular small ones, exhibited vigorous mobility (Supplemental Movie 2 and Fig. 3), which might relate to the roles of full-length huntingtin in axonal transport[2]. We note, however, that we found no evidence for an association of endogenous huntingtin with Htt134:EGFP aggregates (Supplementary Fig. 6). Interestingly, both here (Fig. 2B) and in our prior study[36], plots of aggregate appearance over time appeared to be nearly linear, arguing against a dominance of fragmentation and secondary nucleation processes in mHtt aggregation (as these would give rise to exponential plots, at least initially). We note, however, that prior attempts to tease out the relative importance of particular biophysical phenomena were often inconclusive[21,55], indicating yet again that aggregation is governed by multiple biophysical process whose relative importance may vary in space and time.

As mentioned above, our data are consistent with suggestions that the large mHtt aggregates observable by light microscopy are 'protective'[20,21,23,24,26], in the sense that they sequester harmful forms of mHtt and reduce their cytosolic levels. Indeed, measurements based on automated microscopy revealed that inclusion body formation leads to decreased levels of mutant huntingtin elsewhere in a neuron and predicts improved survival[20,21], whereas observations made in post-mortem tissue of HD patients reported that aggregates are much less common in striatal neurons compared to cortical neurons even though striatal neuron loss is much more pronounced[22,23]. Interestingly, these two studies[22,23] reported that the overwhelming majority of aggregates in postmortem preparations were located in the neuropil, with nuclear aggregates showing up only at late disease stages or in more severe HD presentations (i.e., juvenile). Many neuropil aggregates were observed to reside in dendrites, although, in common with our observations (Fig. 3) many of the processes containing aggregates in the postmortem preparations were not identifiable, due to poor tissue preservation or the lack of distinctive morphology[22]. In our study, we concluded that most of mHtt134:EGFP aggregates were axonal, a conclusion primarily based on tracking about 700 aggregates individually over many time points. Our conclusions might have been different if assignments were based on single time-points, or 'snapshots'.

In contrast to the primarily axonal localization of most mHtt134Q:EGFP aggregates, we only rarely observed aggregates in nuclei, in agreement with the postmortem studies mentioned above[22,23] but in striking contrast to other studies carried out using neuronal cell cultures (e.g., refs. 19,20) which we attribute to our improved experimental conditions discussed above (also compare Figs. 1–3 with Supplementary Fig. 2). This, together with the fact that neurons appeared healthy and survived for the entire duration of these long experiments (nearly 4 weeks) would seem to suggest that extranuclear, and in particular axonal aggregates, residing at locations remote from the cell body, are less harmful to cells and enhance their survival capacity. In continuation of this line of reasoning, mutation of lysines 6 and 9 to arginines, which was associated with pervasive cell death, resulted in much less peripheral sequestration and a widespread appearance of nuclear aggregates. Such aggregates, or inclusion bodies, were previously suggested to be particularly toxic[64–66] and are ultrastructurally distinct[18,19]. Yet as other studies have questioned their association with cell death[22–24,67,68], some caution should be exercised regarding their causal role in cell death presumably related to the elimination of the two aforementioned ubiquitination sites. In spite of these many caveats, however, the most parsimonious interpretation of this study as a whole is that N-terminal mHtt ubiquitination promotes its sequestration into highly concentrated, spatially confined yet dynamic pools at peripheral cellular localizations, and by doing so, reduces the toxic load of cytosolic mHtt forms and prolongs neuronal survival in face of the cytotoxic challenge these mHtt forms pose.

## Materials and methods
### Primary cultures of rat cortical neurons
Experiments were performed in primary cultures of newborn rat neurons (Wistar, either sex) prepared according to a protocol approved by the "Technion, Israel Institute of Technology Committee for the Supervision of Animal Experiments" (Approval IL-105-08-20). These primary cultures of rat cortical neurons were prepared in compliance with all relevant ethical regulations for animal use, as described before[41]. Specifically, cortices of 0- to 1-d-old rats were dissected, dissociated by trypsin, followed by trituration using a siliconized Pasteur pipette. Cells were plated on thin glass multi-electrode array (MEA) dishes (Multichannel Systems), on coverslips, or on glass bottom dishes (MatTek). All plates/chambers were pre-treated with polyethylenimine (Sigma-Aldrich) to facilitate cell attachment. Cells were initially grown in a medium containing Eagle's minimal essential medium (Sigma-Aldrich), 25-µg/mL insulin (Sigma-Aldrich), 20-mM glucose (Sigma-Aldrich), 2-mM L-glutamine (Sigma-Aldrich), 10% NuSerum (Becton Dickinson Labware). The preparation was then transferred to a humidified tissue culture incubator and maintained at 37 °C in a 95% air and 5% $CO_2$ mixture. Half the volume of the culture medium was replaced once a week (coverslips and glass bottom dishes) or three times a week (MEA plates) with feeding media similar to the media described above, but devoid of NuSerum, and with a lower concentration (0.5 mM) of L-glutamine and supplemented with 2% B-27 (Gibco).

### DNA constructs
The DNA construct used to express Htt134Q:EGFP was previously described[36] and was generated as follows: Large scale gene synthesis was used to synthesize the first exon of the human HTT gene containing 134 CAG repeats, fused to EGFP and flanked by AgeI (5′) and BsrGI (3′). This segment was inserted into FUGWm (a modified version of the FUGW 3rd generation lentivirus backbone[69]) upstream to EGFP, using the AgeI and BsrGI sites, to create FUHtt134Q:EGFP-Wm. All gene synthesis and cloning were carried out by GenScript (Piscataway NJ, USA). Lys 6 and Lys 9 in FU-Htt134Q:EGFP-Wm were mutated to Arg residues to create FU-Htt134Q(K > R):EGFP-Wm (please note that this same construct was previously described as FU-Htt134Qm:EGFP-Wm[36]).

The DNA construct used to express Htt134Q:EGFP:T2A:mCherry was generated as follows: The DNA fragment BsrGI-T2A-mCherry-XhoI was synthesized de-novo and inserted into FU-Htt134Q:EGFP-Wm using the restriction enzymes BsrGI and XhoI (downstream to the EGFP segment). The DNA construct used to express 134Q(K > R):EGFP:T2A:mCherry was generated as follows: Lys 6 and Lys 9 in 134Q:EGFP:T2A:mCherry were replaced by Arg residues to create the final plasmid. All cloning and gene synthesis were carried out by GenScript.

The DNA construct used to express Htt134Q:HaloTag was generated as follows: 6 bp (GCTAGC; Ala-Ser) were inserted in frame into FUHtt134Q:EGFP-Wm to create a new NheI restriction site at position 4490 (between the mHtt segment and EGFP). C-term-Halotag-7 flanked by NheI and XhoI (NheI-C-term-Halotag-7-XhoI) was synthesized de novo and inserted using NheI and XhoI, replacing the EGFP segment. The DNA construct used to express 134Q(K > R):HaloTag was generated as follows: Lys 6 and Lys 9 in FUHtt134Q:EGFP-Wm were replaced by Arg residues using site direct mutagenesis, to create the final plasmid. All cloning and gene synthesis were carried out by GenScript.

### Lentivirus production and transduction
Lentiviral particles were produced by transfecting HEK293T cells with a mixture of two plasmids expressing the key HIV packaging genes and a heterologous viral envelope gene (MISSION® Lentiviral Packaging Mix, Sigma). HEK293T cell transfection was performed using Lipofectamine 2000 (Invitrogen) in 10 cm plates in cells that had reached 90% confluence. Supernatant was collected after 48–72 hours, passed through 0.45 µm filters, aliquoted, and stored at −80 °C.

### Long-term imaging
MEA dishes containing cortical neurons and expressing one of the aforementioned mHtt fusion proteins were mounted on a custom-built confocal laser scanning (inverted) microscope based on a Zeiss Axio Observer Z1, using a 40×, 1.3 N.A. Plan-Fluar objective. The system was controlled by custom software and includes provisions for automated, multisite time-lapse microscopy. The dish was then covered with a custom-designed cap

containing inlet and outlet ports for perfusion and air as well as a reference ground electrode. MEA dishes were continuously perfused with feeding media (described above) supplemented with 10% distilled water (to compensate for evaporation) at a rate of 2 ml/d by means of a custom-built perfusion system based on ultraslow flow peristaltic pump (Instech Laboratories, Inc.) and silicone tubing. The tubes were connected to the dish through the appropriate ports in the custom designed cap. A 95% air / 5% $CO_2$ sterile mixture was continuously streamed into the dish at very low rates through a port with flow rates regulated by a high precision flow meter (Gilmont Instruments). The bases of the headstage/amplifier and the objective were heated to 37 °C using resistive elements and separate temperature sensors and controllers, stabilizing a temperature of 35–36 °C in the culture media. Excitation was carried out using 488 nm, 594 nm 632 nm solid state and gas lasers (Coherent; Cobolt, Uniphase). Fluorescence emissions were read through either 500–550 nm band-pass (Chroma Technology), 594 nm long pass (Semrock) or a 633 nm long pass (Semrock) filters. Time-lapse recordings were usually performed by averaging 5 frames at 12 focal planes spaced 0.8 µm apart. All data were collected at a resolution of 640×480 pixels, at 12 bits/pixel. Data were collected sequentially from multiple sites using a motorized stage to cycle automatically through these sites at 90 or 120-min intervals. Focal drift was corrected automatically by using the confocal microscope autofocus system.

### HaloTag ligands
The following Halotag ligands were used in this study: Janelia Fluor 635 (JF635-HT), Janelia Fluor 479 (JF479-HT) and Janelia Fluor 635 impermeable (JF635i-HT) were provided as generous gifts by Luke Lavis, Janelia Research Campus[70]; Oregon Green cell permeable ligand was procured from Promega. All ligands were kept as 100 µM stocks at −20 °C. The non-fluorescent HaloTag blocker CPXH[49] {1-chloro-6-(2-propoxyethoxy) hexane} was synthesized at our request by AKos Consulting & Solutions, GmbH. The dry material was dissolved in DMSO to prepare a 10 mM stock solution, which was stored in small aliquots at −20 °C until used.

### HaloTag pulse chase experiments
Labeling of mHtt-Halotag fusion proteins was carried out as follows: Oregon Green cell permeable ligand or JF479-HT were added to the dishes on the microscope for 1 h at a final concentration of 100 nM, following by gently washing it out (replacing half of the media 4 times). Following washing, CPXH (10 µM) was added for ~4 hours to saturate remaining ligand-free HaloTag protein sites. CPXH was then washed out by replacing half of the media 4 times as described above, followed by the addition of JF635-HT ligand to a final concentration of 100 nM, which was kept in the media for the rest of the experiment. Imaging was initiated immediately after CPXH addition. Data was collected as mentioned above. For the experiments with the impermeable HaloTag ligand (Supplementary Fig. 3) the experimental procedure was as follows: Oregon Green cell permeable ligand was added on the microscope to cells expressing 134Q:HaloTag for ~1 h, after which it was washed as described above and imaging was initiated. One baseline image was collected, and then JF635i-HT was added to a final concentration of 100 nM and left in the media for the rest of the experiment. After collecting two images at 1.5 h min intervals, JF635-HT at a final concentration of 100 nM was added to the dish.

### Fluorescence recovery after photobleaching (FRAP)
Cortical neurons were infected with viral particles encoding for Htt134Q:EGFP at day 12 in culture. 10 days post infection, cells were mounted on the microscope and connected to the perfusion system as mentioned above. mHtt134Q:EGFP positive aggregates were imaged first to obtain baseline fluorescence levels, and then individual fluorescent aggregates were selectively photobleached using 488 nm at maximal laser power and the systems AOTF to limit exposure to selected aggregates. Time lapse imaging was then initiated at 5-minutes intervals

between cycles. Fluorescence recovery of individual aggregates was analyzed as described below.

### Image analysis
All imaging data analysis was performed using custom written software ('OpenView') which includes features for automated/manual tracking of individual objects and measurements of fluorescent intensities over time (described in detail in ref. 42). To measure the fluorescence intensity of aggregates, region of interest (ROIs) were placed programmatically on fluorescent punctate objects at each time step using identical parameters, and mean pixel intensities within these ROIs were obtained from maximal intensity projections of Z section stacks. To measure cytosolic mHtt fluorescence levels, ROIs were placed manually at the first-time step on somatodendritic or axonal areas, and mean pixel intensities within these areas were obtained from maximal intensity projections of Z section stacks for each time step. For analyzing FRAP experiments, individual aggregates were identified and tracked semiautomatically, and mean pixel intensities within these areas were obtained from maximal intensity projections of Z section stacks.

### Immunofluorescence
For the data of Supplementary Fig. 2, cortical neurons were plated on glass coverslips to which 8 mm glass cylinders (Belco glass) were adhered using sterile silicon grease. Approximately $3 \times 10^4$ cells were plated on each cylinder and allowed to develop for 9 days. At DIV 9, viral particles of FU-Htt134Q:EGFP-Wm or FU-Htt134Q(K > R):EGFP-Wm were added to each cylinder. At DIV 19, cells were fixed using either of two protocols: 1. PFA fixation: Cells were washed with Tyrode's physiological solution (119 mM NaCl, 2.5 mM KCl, 2 mM $MgCl_2$, 25 mM HEPES; 30 mM Glucose and 2 mM $CaCl_2$, pH 7.4) and exposed to the fixative solution (4% formaldehyde and 4% Sucrose in PBS) for 20 min at room temperature. This was followed by adding fixative buffer supplemented with Triton X-100 (0.25%) for an additional 20 min. Cells were then washed with PBS. 2. Methanol fixation: Cells were washed with Tyrode's solution, whose level, after the final wash, was reduced as much as possible without exposing the cells. Then, the cells were exposed to 100% cold Methanol (stored at −20 °C), the coverslips were placed on a cold metal block and moved to -20 °C for 15 min. Cells were then washed with PBS. Fixed cells (both protocols) were then incubated with 10% bovine serum albumin (BSA) for 1 h at 37 °C and exposed to the primary monoclonal mouse anti - Neurofilament Heavy chain (Imgenex, clone NAP4, #IMG-5018A-1, 1:1000). The cells were incubated with the antibody for 1 h at 37 °C, washed and then labeled with a secondary antibody (Cy™5 AffiniPure Donkey Anti-Mouse IgG; Jackson ImmunoResearch Laboratories, #715-225-151). The cells were then washed, and stained with monoclonal mouse anti MAP2-antibody (1:1000, Sigma, #M9942) decorated with labeled Fab fragments (Zenon™ Alexa Fluor™ 594 mouse Labeling Kit; ThermoFisher Scientific). Cells were washed again, treated with Hoechst 33342 for nuclear staining (Thermo) and imaged. Labeled cells were excited using a 405-nm, 488-nm, 594-nm, and 633-nm lasers. Fluorescence emissions were read through a 467-490-nm band pass filter (Semrock), 500–550-nm band-pass filter (Chroma Technology), and a 638-nm long pass filter (Chroma). Data were collected as described above.

For the data of Supplementary Fig. 1, cortical neurons were plated on glass bottom petri dishes (MatTek). Approximately $3 \times 10^4$ cells were plated on each dish and allowed to develop for 9 days. At DIV 9, viral particles of FU-Htt134Q:EGFP were added to each cylinder. At DIV 19, the petri dishes were covered with a custom-designed cap containing inlet and outlet ports for perfusion and air, mounted on an experimental setup, provided with a sterile mixture of 5% $CO_2$ / 95% air, continuously perfused with feeding media as described above and heated to 35–36 °C. After two days, the cells were fixed and stained according to the PFA fixation protocol described above. After 1-hour incubation with 10% BSA, the cells were exposed to

anti-Neurofilament Heavy chain, phosphorylated (BioLegend #801602, clone SMI 31, 1:1000). The cells were incubated with the antibody for 1 h at 37 °C, washed and then labeled with a secondary antibody (Cy™5 AffiniPure Donkey Anti-Mouse IgG; Jackson), and imaged as described above. For the data of Supplementary Fig. 6, an anti Huntingtin rabbit monoclonal antibody (Cell Signaling Technology anti Huntingtin (D7F7) XP® Rabbit mAb #5656, 1:200) was used to stain neurons growing on glass bottom petri dishes after perfusion as described above (one experiment) or without a perfusion period (2 experiments). No colocalization of endogenous huntingtin with Htt134Q:EGFP aggregates was observed in any of these experiments.

## Statistics and reproducibility

The term 'independent experiment' implies a separate cell culture preparation, transduction, dish and imaging session. Data for separate experiments are shown separately. Numbers of individual experiments, fields of views (each containing at least one neuron) and aggregates are provided explicitly in main text, figure legends and figures. Error bars indicate SEM.

## Reporting summary

Further information on research design is available in the Nature Portfolio Reporting Summary linked to this article.

## Data availability

All data, expression vectors and software used in this study are available upon request. Digital data underlying all graphs and plots are provided as Supplemental Data.

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

## Acknowledgements

We are grateful to Tamar Galateanu, Leonid Odesski and for their invaluable assistance, to Huu Phuc Nguyen (Ruhr University, Bochum Germany) and Dr. Libo Yu-Täger (Institute of Medical Genetics and Applied Genomics, University of Tuebingen) for many fruitful discussions, and to all members of the Ziv lab for their support and assistance. The project was supported by grants from the Germany-Israeli Foundation for Scientific Research and Development (GIF; I-1437-418.13/2017) to AC and NEZ, the German Research Foundation (DFG, Research unit 5228, "Syntophagy") to NEZ, the Rappaport Institute and the Allen and Jewel Prince Center for Neurodegenerative Disorders of the Brain. AC is supported by grants from the Adelson Medical Research Foundation (AMRF), the Israel Science Foundation (ISF), A professorship from the Israel Cancer Research Fund (ICRF, USA), the Technion-University of Michigan at Ann Arbor Collaborative Program, and a generous donation by Craig Darian and the late Albert Sweet.

## Author contributions

Project conceptualization: A.B., A.C., and N.E.Z; Experimentation: A.B. and D.M.; Data analyses: A.B., D.M., and N.E.Z.; Funding acquisition: A.C. and N.E.Z; Writing and Method development: A.B. and N.E.Z.

## Competing interests

The authors declare no competing interests.
