## [Peer Review file · Communications Biology]

Peripheral sequestration of huntingtin delays neuronal death and depends on N-terminal ubiquitination

Corresponding Author: Professor Noam Ziv

Version 0:

Reviewer comments:

Reviewer #1

(Remarks to the Author)

Following initial studies on the 'protective' role of inclusion body formation in neurons, and more recent studies on the N-terminal ubiquitination of mutant HTT exon1 also showing that impairing ubiquitination affects toxicity in cortical cell models, the submitted manuscript by Boulos and colleagues explores in more detail the aggregation in long-term experiments. Overall the microscopy studies are of high quality yet the mechanistic insights are lacking (eg affected transport, nuclear localization).

Using a fluorescent pulse-chase experiment a slow exchange of HTT is observed which is not seen by FRAP experiments that are indeed too short to detect an on/off rate, as published before by various groups in cell and c elegans models. The aggregates are not stained using a membrane impermeable fluorescent HaloTag ligand -yet is such a ligand able to label aggregates or can this only happen by sequestering labeled, soluble HTT?
mHTT exon1 is shown to form aggregates in the periphery of cells, not the cell body or neurons but presumably axons. Ideally this should be proven using axonal markers as used for the experiment shown in suppl fig 1 (eg MAP2, NFL) by following cells in time to track aggregates and fix&stain with antibodies.

Mutation of K6K9 reduces aggregate formation in the periphery and the pulse-chase experiment shows reduction of the old pool and stable levels of the newly synthesized pool. The authors suggest that this is due to either reduces exchange or to impaired protein synthesis, but could it be that there is an on/off rate with the old pool being degraded in time and the new pool being degraded but also replaced by newly-synthesized HTT, as the red dye is not washed away?

The observation of nuclear aggregates with the lysine mutant HTT is certainly of interest, yet the discussion mentions that the axonal HTT aggregates may be due to the role of HTT in axonal transport, but this is full-length HTT, not exon1 as being used in the study. Is full-length HTT sequestered in these aggregates too?

Reviewer #2

(Remarks to the Author)

This manuscript by Boulos et al is interesting and provided an in depth, long term imaging of the Htt aggregates and their logistics around the primary cortical neurons of the mice. However, there are some concerns that need to be corrected before publication in any journal.

1. Authors have built their first conclusion of the Htt aggregates starts to form once the cytosolic concentration is reached a certain critical level. Is there any experimental support for this concentration that builds over slowly and there is critical concentration point below which aggregates are not formed but once it is achieved, aggregates are formed and shipped to the axons.
2. This also leads to the second question, why aggregates are shipped to Axon? Any mechanistic explaining for their shipment to the axons?
3. In the cytosolic exchange experiment, the authors conclude that cellular production of Htt is not reduced after formation of Htt aggregates, instead they are Htt is exchanged with the peripheral neurons. This raise a lot of questions. Usually the protein aggregates are formed due to misfolding and tight packing/ β sheets stacking of the protein and once formed, the pathological aggregates are irreversibly formed and do not give away monomers. Also, it is understandable the production of new Htt will not be reduced to cellular programming with the vector, not all of the newly produced Htt will be in equilibrium with the aggregates axonal Htt. Only a limited small amount can achieve this exchange, the rest of new batches of the Htt can just increase the number of aggregates and result in aggregates overload and probably toxicity. This need to be

explained mechanistically.

4. Mutation at K6 and 9 disrupt the ubiquitination of the Htt and thus the aggregates formation but increased the cell death. The aggregated form of the Htt is associated with cell death that is not observed by the authors, but disaggregated form of Htt (due to preclusion of ubiquitination) the cells are dying. This is opposite to what is observed in protein aggregation science.

Reviewer #3

(Remarks to the Author)

In this manuscript, Boulos et.al., describe the kinetics of Htt aggregates using longitudinal imaging with rat cortical neurons. Their findings can be summarized as:

- 1- Exon1 Htt-GFP (expanded Q134) aggregates over time and is sequestered over weeks at the periphery of the neurons (especially axonal aggregates), acting as a “sink” for the soluble forms of the fragments.
- 2- This phenomenon is not seen when two lysines are mutated in N-terminus of Htt Exon1 (These post-translational modifications were found in a previous publication by the same labs). These fragments stay rather diffuse, sometimes forming nuclear aggregates, concomitant with heavy neuronal loss (further extending the original findings in Hakim-Eshed et.al., 2020).
- 3- Halo-tag labeled Htt-Exon1 experiments show that old aggregates can recruit new fragments.
- 4- I found the differences in aggregate behaviors between continuous culture vs infrequent media-change-culture quite intriguing. If true, this is an important consideration for all of us who deal with proteinopathies.

In general, I found the paper quite well written (albeit sometimes verbose), and its advantages/limitations well-articulated. The concept of “large aggregates as protective sinks” is not a new one but it is always debated in the context of neurodegeneration. The results acquired in this paper could provide insights into other proteinopathies, where this debate is still unresolved.

Overall, I see no reason against publication of the paper. Yet, some improvements are necessary for the data representation. Here I enumerate my suggestions.

1- Slightly modifying the construct nomenclature. I found spurious use of the term mHtt-EGFP both in the text, in the figures and figure legends. This is slightly confusing considering the use of Httm-EGFP (representing the K>R mutants). I would recommend dropping the ‘m’ and only use Htt134Q-EGFP for expanded Exon1 fragment. Also in the text Htt134Q is sometimes mentioned as mHtt134Q.

A- Htt134Q:EGFP: Representing the expanded Exon1 but not K>R mutated.

B- Htt134Q(K>R):EGFP: For representing the K>R mutant.

I understand the mHtt represents the expansion, but the extra mutations might confuse the readers on a first read. It certainly confused me.

2- Clearly define replicates. What is the “experiment1 vs 2 vs 3”? Is this a biological replicate where a new cortical culture is started, and new transductions are performed? If so, it would always be good to keep the data points separate and show experimental variability among the replicates.

3- 2a cleavage with mCherry is clever to provide contour for the cell for live cell imaging. Beware that although this cleavage is efficient, it is not perfect and there is always a fusion protein remains. We cannot know the implications of this fusion protein. Especially for protein aggregates where small seeds can trigger aggregations.

3- Would it be possible to co-transduce the cells with an axonal marker in the live cell imaging? This would avoid unlabeled or lowly labeled axons, further convincing the readers.

4- Always use color-blind friendly formats for image representations.

5- I think Figure1-2 can be merged. This is also true for other figures where one figure is the quantification of the other. It might be easier for the reader to look at them side by side.

6- In Figure 5B-E, the Y axis is labeled as F/F0. Please explain in the figure legend what this measure represents. I am assuming this is Fluorescence/ Time0 fluorescence but still maybe label more clearly.

7- The Htt134Qm (K>R) Halo tag pulse experiments do indeed come with the caveat that the second labeling is much less efficient than the first labeling, suggesting a translational defect. However, this caveat is articulated in the paper.

8- I would avoid the term “very long-term microscopy”. “Very” seems unnecessary.

Author Rebuttal letter:

We thank all three reviewers for the meticulous and helpful evaluation of our manuscript. Responses to all points are provided below.

Reviewer #1

Following initial studies on the 'protective' role of inclusion body formation in neurons, and more recent studies on the N-terminal ubiquitination of mutant HTT exon1 also showing that impairing ubiquitination affects toxicity in cortical cell models, the submitted manuscript by Boulos and colleagues explores in more detail the aggregation in long-term experiments. Overall the microscopy studies are of high quality yet the mechanistic insights are lacking (eg affected transport, nuclear localization).

Using a fluorescent pulse-chase experiment a slow exchange of HTT is observed which is not seen by FRAP experiments that are indeed too short to detect an on/off rate, as published before by various groups in cell and *C. elegans* models.

We thank the reviewer for these excellent comments. We acknowledge the lack of molecular mechanistic insights, but wish to stress that our goal in this study was to examine the detailed aggregation process, its impact on cytosolic mHtt levels and cell viability - in the same neurons – and over an extended time window. There are now countless, mechanistic studies on the biophysics and biochemistry of mHtt aggregation (many of which carried out *in vitro* or in non-neuronal cells), yet we are not aware of any study - including the fantastic, seminal papers from the Finkbeiner lab - that documented the aggregation process and the molecular dynamics of individual mHtt aggregates in the same neurons, axons and dendrites, in optimized environmental conditions, at such high temporal and spatial resolutions and for such long durations. We feel that there is considerable merit in knowing what happens in addition to fully understanding the molecular details of how it happens.

1) The aggregates are not stained using a membrane impermeable fluorescent HaloTag ligand -yet is such a ligand able to label aggregates or can this only happen by sequestering labeled, soluble HTT?

The experiments using a membrane impermeable ligand had two parts: First, a membrane impermeable ligand was used and indeed, practically no binding was observed. But then, a membrane permeable ligand was added. As shown in Fig. S3, the membrane permeable ligand rapidly labeled the very same aggregates, that is, the same aggregates that the membrane impermeable ligand failed to label.

More generally, all experiments with mHtt-HaloTag fusion proteins started with the labeling of the preexisting mHtt pool, using a 1-hour exposure to the first HaloTag ligand. As evident in Figs. 4, 5, 9, S3, this resulted in rapid aggregate labeling. Given the slow kinetics of aggregate formation (time constant \approx 48 hours; Fig. 5E; see also Hakim-Eshed et al., 2020, Fig. 2F), it is highly unlikely that these labeled aggregates formed from soluble mHtt within the one hour labeling period.

These observations thus suggest that Htt134Q:HaloTag within aggregates can bind HaloTag ligands directly.

2) Ideally this [axonal localization] should be proven using axonal markers as used for the experiment shown in suppl fig 1 (eg MAP2, NFL) by following cells in time to track aggregates and fix&stain with antibodies.

The live imaging experiments described in the original submission were carried out using neurons grown on reusable thin-glass, multielectrode array (MEA) dishes. Over the years, we developed 'life support' systems (including slow perfusion systems) for such dishes that allow on-microscope, continuous imaging over many days and weeks. Unfortunately, the current cost of each MEA dish is \sim 700 Euro, which challenges our capacity to expose these to damaging fixatives. Consequently, over the last months, we fabricated the necessary parts and adapted our perfusion and life support systems for standard glass-bottomed petri dishes (while this may sound simple, it involved much trial and error to develop parts that ensure contamination-free, long term experiments – please note that we do not add antibiotics to our cell culture media). We then used these to carry out the requested staining with an axonal marker (anti-Neurofilament H, Phosphorylated, BioLegend) in networks experiencing the same environmental conditions. As shown in Fig. S1, the vast majority (\sim 83%) of Htt134Q-EGFP aggregates colocalized with anti-NF-H labeled axons, in line with the predominantly axonal localization deduced from the less-than-ideal mCherry labeling.

3) Mutation of K6K9 reduces aggregate formation in the periphery and the pulse-chase experiment shows reduction of the old pool and stable levels of the newly synthesized pool. The authors suggest that this is due to either reduces exchange or to impaired protein synthesis, but could it be that there is an on/off rate with the old pool being degraded in time and the new pool being degraded but also replaced by newly-synthesized HTT, as the red dye is not washed away?

There is a misunderstanding concerning the "stable levels of the newly synthesized pool". The stable

F/F0 values of the JF635HT (red) channel are not stable because levels of newly synthesized Htt in these aggregates remain stable, but because no labeling whatsoever occurred (the fluorescence remained at baseline background levels). This is now explained in the text (line 317). Therefore, the suggested explanation, while interesting, is not congruent with the observations.

4) The observation of nuclear aggregates with the lysine mutant HTT is certainly of interest, yet the discussion mentions that the axonal HTT aggregates may be due to the role of HTT in axonal transport, but this is full-length HTT, not exon1 as being used in the study. Is full-length HTT sequestered in these aggregates too?

Actually, we did not claim that “axonal HTT aggregates may be due to the role of HTT in axonal transport”; we merely speculated that once mHtt is in axons, the probability for collision-induced aggregate formation might increase due to the tightly constrained axonal volume, on the one hand, and the presence of extensive transport dynamics within axons, on the other. To avoid this misunderstanding, we modified the text (lines 442-449).

As to whether full length Htt is sequestered in the same aggregates – as far as we can tell, the answer is No. In response to the reviewer’s suggestion, and after consultation with our colleagues in Tuebingen, we procured a monoclonal antibody prepared against a Htt domain not encoded by exon 1 (a synthetic peptide corresponding to residues surrounding Pro1218 of human huntingtin protein; Cell Signaling Technology Huntingtin (D7F7) XP® Rabbit mAb #5656). Using this Ab, we stained preparations containing multiple Htt134Q-EGFP aggregates, but as shown in a new Figure (Supp. Fig. 6), we observed absolutely no colocalization of these aggregates with endogenous huntingtin. This observation is now also mentioned in the main text (lines 447-449).

Reviewer #2

This manuscript by Boulos et al is interesting and provided an in depth, long term imaging of the Htt aggregates and their logistics around the primary cortical neurons of the mice. However, there are some concerns that need to be corrected before publication in any journal.

We thank the reviewer for this appraisal.

1) Authors have built their first conclusion of the Htt aggregates starts to form once the cytosolic concentration is reached a certain critical level. Is there any experimental support for this concentration that builds over slowly and there is critical concentration point below which aggregates are not formed but once it is achieved, aggregates are formed and shipped to the axons.

If we understand correctly, the reviewer is asking whether experimental evidence was obtained that supports two separate conclusions: a) the existence of critical concentrations, and b) the shipping of aggregates to axons. We address these separately.

a) Critical concentrations - the time course of changes in cytosolic mHtt134Q:EGFP fluorescence levels reveals a consistent pattern – a gradual increase in cytosolic mHtt134Q:EGFP fluorescence followed by a reduction in this fluorescence. More or less at the same time, mHtt134Q:EGFP aggregate counts start rising. While these data are congruent with the notion of a critical concentration (e.g. Pei et al., 2021), the reviewer is correct in questioning whether there is experimental support for this conclusion. We note that we have not claimed (nor do we now) that our data provide conclusive evidence for the existence of critical concentrations. We only offered this as a possible interpretation of the findings of Figs 1 and 2. To remove all doubt, however, we removed any mention of critical concentrations from the Results. We wish to stress that the matter of critical concentrations was not the main point of these experiments. Rather, the main point was the apparent efficacy of (peripheral) aggregates in sequestering away cytosolic mHtt134Q:EGFP.

b) Shipping of aggregates to axons – we have not claimed in any way that aggregates are “shipped to axons”. To the contrary - our data suggest that they are formed ab initio in the cell periphery, mainly in axons (please see our next reply and the reply to the last comment made by Reviewer #1).

2) This also leads to the second question, why aggregates are shipped to Axon? Any mechanistic explaining for their shipment to the axons?

We are not claiming that aggregates are “shipped to axons”; rather our data suggests that they form ab initio in the neuronal periphery, primarily in axons, and possibly also in thin, remote dendrites (in excellent agreement with post-mortem observations in HD patients, e.g. Gutekunst et al., 1999; Kuemmerle et al., 1999). Once formed, however, some aggregates definitely seem to move along thin neurites, presumably axons.

We strongly recommend to view the supplemental movies – these provide no evidence for a stream of aggregates moving out of the cell body; conversely, they do illustrate numerous events of aggregates moving abruptly in straight lines, presumably along axons or very thin dendrites. We do not know for what underlies these erratic and occasional movements but the underlying processes responsible for these events are not the main point; rather, the point is that aggregates tend to form at the cell's periphery, in thin neurites, presumably axons, and we took advantage of the linear movements some aggregates exhibit as evidence that these reside within hard-to-resolve peripheral neurites.

3) In the cytosolic exchange experiment, the authors conclude that cellular production of Htt is not reduced after formation of Htt aggregates, instead they are [claiming] that Htt is exchanged with the peripheral neurons [should be aggregates?]. This raises a lot of questions. Usually the protein aggregates are formed due to misfolding and tight packing/ β sheets stacking of the protein and once formed, the pathological aggregates are irreversibly formed and do not give away monomers.

Regarding the irreversibility of mHtt binding, several studies have shown already that this is not entirely correct. For example, the ability of aggregates to exchange mHtt with cytosolic pools has already been reported in yeast and cell lines (Peskett et al., 2018; Aktar et al., 2019; Pei et al., 2021) using fluorescence recovery after photobleaching (FRAP), photoconversion and rapid depletion of cytosolic mHtt. Our data are in line with these findings but indicate that in neurons, exchange rates are much slower than those reported in yeast. Yet, slow as these may be, exchange with cytosolic pools is clearly observed when the observation window is expanded from minutes/hours (prior studies) to many days (current manuscript, Figs. 4, 5 and S4). In fact, extended observation windows have provided some evidence for reversible aggregate formation in neurons as early as 2004 in the seminal paper of the Finkbeiner group (Arrasate et al., 2004). Therefore, in our humble opinion, the term 'irreversible' used to describe aggregates might need some reconsideration.

Also, it is understandable the production of new Htt will not be reduced to cellular programming with the vector, not all of the newly produced Htt will be in equilibrium with the aggregates axonal Htt. Only a limited small amount can achieve this exchange, the rest of new batches of the Htt can just increase the number of aggregates and result in aggregates overload and probably toxicity. This needs to be explained mechanistically.

As indicated by the reviewer, as more and more mHtt is synthesized, the capacity of a fixed aggregate pool to buffer cytosolic mHtt will reach saturation. However, the aggregate number is not fixed either. In fact, and as suggested by the reviewer, we do see gradual increases in the number of aggregates (Fig. 2B; please also see Hakim-Eshed et al., 2020, Fig. 2) which is strongly suggestive of a gradually increasing buffering capacity. Interestingly, cytosolic levels continue to diminish, which is not congruent with a saturation of aggregate buffering capacity (although, of course, other processes might 'kick in', such as a suppression of protein synthesis, elevated degradation, etc.). It is highly likely that with time, the neurons will succumb to aggregate overload, but in the time frame of these experiments (about a month), we did not observe this to occur. In contrast, when the KK>RR variants were expressed, less aggregates formed and cell death was widespread within this same time frame. This being so, we feel that the point raised by the reviewer is in full agreement with the findings and well-explained by them.

4) Mutation at K6 and 9 disrupt the ubiquitination of the Htt and thus the aggregates formation but increased the cell death. The aggregated form of the Htt is associated with cell death that is not observed by the authors, but disaggregated form of Htt (due to preclusion of ubiquitination) the cells are dying. This is opposite to what is observed in protein aggregation science.

This indeed seems puzzling at first sight. However, we previously reported (Hakim-Eshed et al., 2020) that while these particular mutations result in less large aggregates (that is, aggregates discernable by light microscopy), they paradoxically result in more insoluble mHtt as detected using biochemical methods, in particular filter traps. Moreover, these mutations were found to impair cell viability when expressed in cell lines (please see Hakim-Eshed et al., 2020, in particular Figs. 3 and 4). A large body of evidence suggests that small mHtt oligomers rather than large discernable aggregates are the most toxic forms of mHtt (please see Ziv and Ciechanover, 2021 for a short review). Our prior and current findings are in full agreement with this evidence.

Reviewer #3

In this manuscript, Boulos et al., describe the kinetics of Htt aggregates using longitudinal imaging with rat cortical neurons. Their findings can be summarized as:

1- Exon1 Htt-GFP (expanded Q134) aggregates over time and is sequestered over weeks at the periphery of the neurons (especially axonal aggregates), acting as a "sink" for the soluble forms of the fragments.

2- This phenomenon is not seen when two lysines are mutated in N-terminus of Htt Exon1 (These post-translational modifications were found in a previous publication by the same labs). These fragments stay rather diffuse, sometimes forming nuclear aggregates, concomitant with heavy neuronal loss (further extending the original findings in Hakim-Eshed et.al., 2020).

3- Halo-tag labeled Htt-Exon1 experiments show that old aggregates can recruit new fragments.

4- I found the differences in aggregate behaviors between continuous culture vs infrequent media-change-culture quite intriguing. If true, this is an important consideration for all of us who deal with proteinopathies.

In general, I found the paper quite well written (albeit sometimes verbose), and its advantages/limitations well-articulated. The concept of "large aggregates as protective sinks" is not a new one but it is always debated in the context of neurodegeneration. The results acquired in this paper could provide insights into other proteinopathies, where this debate is still unresolved.

Overall, I see no reason against publication of the paper. Yet, some improvements are necessary for the data representation. Here I enumerate my suggestions.

We thank the reviewer for this appraisal and the suggestions.

1- Slightly modifying the construct nomenclature. I found spurious use of the term mHtt-EGFP both in the text, in the figures and figure legends. This is slightly confusing considering the use of Httm-EGFP (representing the K>R mutants). I would recommend dropping the 'm' and only use Htt134Q-EGFP for expanded Exon1 fragment. Also in the text Htt134Q is sometimes mentioned as mHtt134Q.

A- Htt134Q:EGFP: Representing the expanded Exon1 but not K>R mutated.

B- Htt134Q(K>R):EGFP: For representing the K>R mutant.

I understand the mHtt represents the expansion, but the extra mutations might confuse the readers on a first read. It certainly confused me.

We apologize for the confusion and thank the reviewer for this suggestion. Indeed, the nomenclature was confusing and inconsistent. The text and figures were modified throughout the manuscript and figures as suggested.

2. Clearly define replicates. What is the "experiment1 vs 2 vs 3"? Is this a biological replicate where a new cortical culture is started, and new transductions are performed?

Yes, exactly. In addition, each independent experiment includes multiple cells (equivalent to Fields of View, or FOVs listed in figures) and of course aggregates (many hundreds). We now added a definition of "independent experiments" as "separate cell culture preparations and transductions" in the Results (lines 145-146).

If so, it would always be good to keep the data points separate and show experimental variability among the replicates.

This is exactly what we did (please see Figs. 2, 3, 5, 7, 8, 9, S1).

3. Cleavage with mCherry is clever to provide contour for the cell for live cell imaging. Beware that although this cleavage is efficient, it is not perfect and there is always a fusion protein remains. We cannot know the implications of this fusion protein. Especially for protein aggregates where small seeds can trigger aggregations.

Indeed. Please note, however, that comparable observations regarding mHtt aggregation and the effects of KK>RR mutations were obtained with GFP tags alone (that is, without the mCherry addition) in Hakim-Eshed et al., 2020 and with a HaloTag reporter (current report). Nevertheless, we now raise this cautionary note in the discussion (lines 377-379).

3 [should be 4]. Would it be possible to co-transduce the cells with an axonal marker in the live cell imaging? This would avoid unlabeled or lowly labeled axons, further convincing the readers.

Following this suggestion and a suggestion made by Reviewer #1, we attempted to co-transduce neurons expressing Htt134Q:EGFP with a second virus expressing myristolated mTurquoise2. While this provided excellent axonal labeling all the way to their farthest reaches, the fraction of co-expressing neurons was not sufficiently high, and thus the challenge to assign many of the Htt134Q:EGFP aggregates to particular neuronal compartments was not satisfactorily resolved.

Instead, we developed the tools to fix and immunolabel neurons maintained in the improved cell culture conditions, using an axonal marker as described in our reply to Reviewer #1 comment 2 (see also new Supp. Figure 1)

4. [should be 5]. Always use color-blind friendly formats for image representations.

All Red/Green panels were replaced with Magenta/Yellow panels.

5 [should be 6]. I think Figure 1-2 can be merged. This is also true for other figures where one figure is the quantification of the other. It might be easier for the reader to look at them side by side.

Merging these panels makes sense; yet the penalty is reducing the size of the time lapse images to the point that it will be difficult to discern details in the panels of Fig. 1C. (in particular, the small, punctate aggregates). We thus prefer to keep these figures separate.

6 [should be 7]. In Figure 5B-E, the Y axis is labeled as F/F₀. Please explain in the figure legend what this measure represents. I am assuming this is Fluorescence/ Time₀ fluorescence but still maybe label more clearly.

Thank you – the Y axis label was modified so this is clearer. The same correction was applied to Fig. 9.

7 [should be 8]. The Htt134Qm (K>R) Halo tag pulse experiments do indeed come with the caveat that the second labeling is much less efficient than the first labeling, suggesting a translational defect. However, this caveat is articulated in the paper.

This is correct and indeed is discussed at some length (lines 319-328). It is worth noting that if these mutations lead to translational defects, the conclusion that the K>R mutations have a negative impact on cell viability is supported further.

8 [should be 9]. I would avoid the term “very long-term microscopy”. “Very” seems unnecessary.

We thank the reviewer for the suggestion. The term was avoided as suggested.

References

- Aktar F, Burudpakdee C, Polanco M, Pei S, Swayne TC, Lipke PN, Emtage L. (2019) The huntingtin inclusion is a dynamic phase-separated compartment. *Life Sci Alliance*. 2:e201900489.
- Arrasate M, Mitra S, Schweitzer ES, Segal MR, Finkbeiner S. (2004) Inclusion body formation reduces levels of mutant huntingtin and the risk of neuronal death. *Nature*. 431:805-10.
- Gutkunst CA, Li SH, Yi H, Mulroy JS, Kuemmerle S, Jones R, Rye D, Ferrante RJ, Hersch SM, Li XJ. (1999) Nuclear and neuropil aggregates in Huntington's disease: relationship to neuropathology. *J Neurosci* 19:2522-34.
- Hakim-Eshed V, Boulos A, Cohen-Rosenzweig C, Yu-Taeger L, Ziv T, Kwon YT, Riess O, Phuc Nguyen HH, Ziv NE, Ciechanover A. (2020) Site-specific ubiquitination of pathogenic huntingtin attenuates its deleterious effects. *Proc Natl Acad Sci U S A*. 117:18661-18669.
- Kuemmerle S, Gutkunst CA, Klein AM, Li XJ, Li SH, Beal MF, Hersch SM, Ferrante RJ. (1999) Huntington aggregates may not predict neuronal death in Huntington's disease. *Ann Neurol*. 46:842-9.
- Pei S, Swayne TC, Morris JF, Emtage L. (2021) Threshold concentration and random collision determine the growth of the huntingtin inclusion from a stable core. *Commun Biol*. 4:971.
- Peskett TR, Rau F, O'Driscoll J, Patani R, Lowe AR, Saibil HR. (2018) A Liquid to Solid Phase Transition Underlying Pathological Huntingtin Exon1 Aggregation. *Mol Cell*. 70:588-601.

Version 1:

Reviewer comments:

Reviewer #1

(Remarks to the Author)

The authors have addressed the raised questions including new stainings (I underscore the amount of work that was needed here, much appreciated!)

Author Rebuttal letter:

REVIEWERS' COMMENTS:

Reviewer #1 (Remarks to the Author):

The authors have addressed the raised questions including new stainings (I underscore the amount of work that was needed here, much appreciated!)

We thank all three reviewers for their constructive reviews.
